# Visual Generation Without Guidance

**Huayu Chen\* [1]   Kai Jiang\* [1 2]   Kaiwen Zheng [1]   Jianfei Chen [1]   Hang Su [1]   Jun Zhu [1 2]**

## Abstract

Classifier-Free Guidance (CFG) has been a default technique in various visual generative models, yet it requires inference from both conditional and unconditional models during sampling. We propose to build visual models that are free from guided sampling. The resulting algorithm, Guidance-Free Training (GFT), matches the performance of CFG while reducing sampling to a single model, halving the computational cost. Unlike previous distillation-based approaches that rely on pretrained CFG networks, GFT enables training directly from scratch. GFT is simple to implement. It retains the same maximum likelihood objective as CFG and differs mainly in the parameterization of conditional models. Implementing GFT requires only minimal modifications to existing codebases, as most design choices and hyperparameters are directly inherited from CFG. Our extensive experiments across five distinct visual models demonstrate the effectiveness and versatility of GFT. Across domains of diffusion, autoregressive, and masked-prediction modeling, GFT consistently achieves comparable or even lower FID scores, with similar diversity-fidelity trade-offs compared with CFG baselines, all while being guidance-free. Code: https://github.com/thu-ml/GFT.

## 1. Introduction

Low-temperature sampling is a critical technique for enhancing generation quality by focusing only on the model's high-likelihood areas. Visual models mainly achieve this via Classifier-Free Guidance (CFG) (Ho & Salimans, 2022). As illustrated in Fig. 1 (left), CFG jointly optimizes the target conditional model and an extra unconditional model during training, and combines them to define the sampling process.

---

\*Equal contribution [1]Department of Computer Science & Technology, Tsinghua University [2]ShengShu, Beijing, China. Correspondence to: Jun Zhu <dcszj@tsinghua.edu.cn>.

*Proceedings of the 42$^{nd}$ International Conference on Machine Learning*, Vancouver, Canada. PMLR 267, 2025. Copyright 2025 by the author(s).

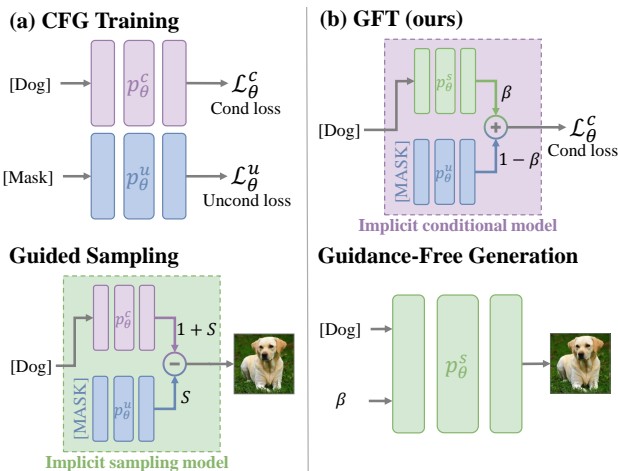

Figure 1: Comparison of GFT and CFG method. GFT shares CFG's training objective but has a different parameterization technique for the conditional model. This enables direct training of an explicit sampling model.

By altering the guidance scale $s$, it can flexibly trade off image fidelity and diversity, while significantly improving the sample quality. Due to its effectiveness, CFG has been adopted as a default technique for a wide spectrum of visual generative models, including diffusion (Ho et al., 2020), autoregressive (AR) (Chen et al., 2020; Tian et al., 2024), and masked-prediction models (Chang et al., 2022; Li et al., 2023).

However, CFG is not problemless. First, the reliance on an extra unconditional model doubles the sampling cost compared with a vanilla conditional generation. Second, this reliance complicates the post-training of visual models – when distilling pretrained diffusion models (Meng et al., 2023; Luo et al., 2023; Yin et al., 2024) for fast inference or applying RLHF techniques (Black et al., 2023; Chen et al., 2024b), the extra unconditional model needs to be specially considered in the algorithm design. Third, this also renders a sharp difference from low-temperature sampling in language models (LMs), where a single model is sufficient to represent the sampling distributions across various temperatures. Similarly following LMs' approach to divide model output by a constant temperature value is generally ineffective in visual sampling (Dhariwal & Nichol, 2021), even for visual AR models with similar architecture to LMs

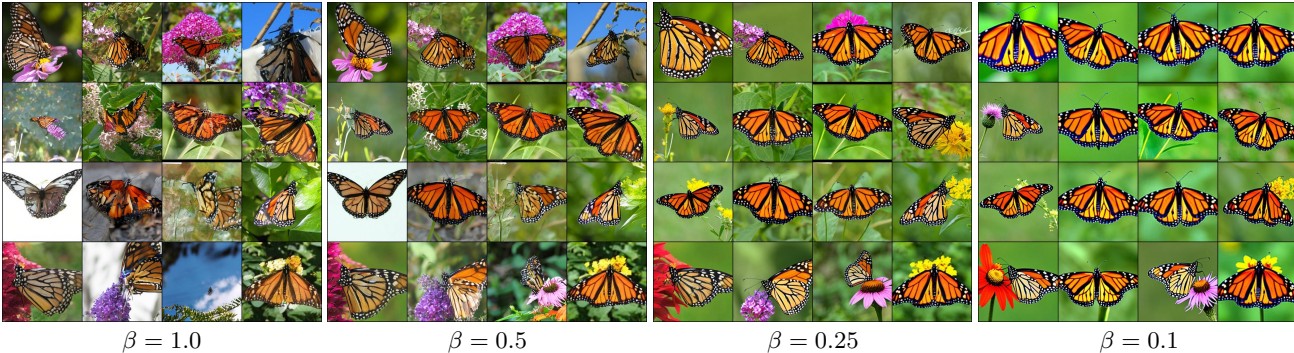

| $\beta = 1.0$ | $\beta = 0.5$ | $\beta = 0.25$ | $\beta = 0.1$ |

Figure 2: Impact of adjusting GFT sampling temperature $\beta$ for guidance-free DiT-XL/2. GFT achieves similar results to CFG without requiring dual model inference at each step. More examples are in Figure 13.

(Sun et al., 2024). All these lead us to ask, *can we effectively control the sampling temperature for visual models using one single model?*

Existing attempts like distillation methods for diffusion models (Meng et al., 2023; Luo et al., 2023; Yin et al., 2024) and alignment methods for AR models (Chen et al., 2024b) are not ultimate solutions. They all rely heavily on pretrained CFG networks for loss definition and do not support training guidance-free models from scratch. Their two-stage optimization pipeline may also lead to performance loss compared with CFG, even after extensive tuning. Generalizability is also a concern. Current methods are typically tailored for either continuous diffusion models or discrete AR models, lacking the versatility to cover all domains.

We propose Guidance-Free Training (GFT), a foundational algorithm for building visual generative models with no guidance. GFT matches CFG in performance while requiring only a single model for temperature-controlled sampling, effectively halving sampling costs compared with CFG. GFT offers stable and efficient training with the same convergence rate as CFG, almost no extra memory usage, and only 10–20% additional computation per training update. GFT is highly versatile, applicable in all visual domains within CFG's scope, including diffusion, AR, and masked models.

The core idea behind GFT is to transform the desired sampling model into easily learnable forms. GFT optimizes the same conditional objective as CFG. However, instead of aiming to learn an explicit conditional network, GFT defines the conditional model implicitly as the linear interpolation of a sampling network and the unconditional network (Figure 1). By training this *implicit* model, GFT directly optimizes the underlying sampling network, which is then employed for visual generation without guidance. In essence, one can consider GFT simply as a conditional parameterization technique in CFG training. This perspective makes GFT extremely easy to implement based on existing codebases, requiring only a few lines of modifications and

with most design choices and hyperparameters inherited.

We verify the effectiveness and efficiency of GFT in both class-to-image and text-to-image tasks, spanning 5 distinctive types of visual models: DiT (Peebles & Xie, 2023), VAR (Tian et al., 2024), LlamaGen (Sun et al., 2024), MAR (Li et al., 2024) and LDM (Rombach et al., 2022). Across all models, GFT enjoys almost lossless FID in fine-tuning existing CFG models into guidance-free models (Sec. 5.2). For instance, we achieve a guidance-free FID of 1.99 for the DiT-XL model with only 2% of pretraining epochs, while the CFG performance is 2.11. This surpasses previous distillation and alignment methods in their respective application domains. GFT also demonstrates great superiority in building guidance-free models from scratch. With the same amount of training epochs, GFT models generally match or even outperform CFG models, despite being 50% cheaper in sampling (Sec. 5.3). By taking in a temperature parameter as model input, GFT can achieve a flexible diversity-fidelity trade-off similar to CFG (Sec. 5.4).

## 2. Background

### 2.1. Visual Generative Modeling

**Continuous diffusion models.** Diffusion models (Ho et al., 2020) define a forward process that gradually injects noises into clean images from data distribution $p(\boldsymbol{x})$:

$$\boldsymbol{x}_t = \alpha_t \boldsymbol{x} + \sigma_t \boldsymbol{\epsilon},$$

where $t \in [0, 1]$, and $\boldsymbol{\epsilon}$ is standard Gaussian noise. $\alpha_t, \sigma_t$ defines the denoising schedule. We have

$$p_t(\boldsymbol{x}_t) = \int \mathcal{N}(\boldsymbol{x}_t | \alpha_t \boldsymbol{x}, \sigma_t^2 \boldsymbol{I}) p(\boldsymbol{x}) \mathrm{d}\boldsymbol{x},$$

where $p_0(\boldsymbol{x}) = p(\boldsymbol{x})$ and $p_1 \approx \mathcal{N}(0, 1)$.

Given data following $p(\boldsymbol{x}, \boldsymbol{c})$, we can train conditional diffusion models by predicting the Gaussian noise added to $\boldsymbol{x}_t$.

$$\min_{\theta} \mathbb{E}_{p(\boldsymbol{x}, \boldsymbol{c}), t, \boldsymbol{\epsilon}} \left[ \| \epsilon_\theta(\boldsymbol{x}_t | \boldsymbol{c}) - \boldsymbol{\epsilon} \|_2^2 \right]. \quad (1)$$

More formally, Song et al. (2021) proved that Eq. (1) is essentially performing maximum likelihood training with evidence lower bound (ELBO). Also, the denoising model $\epsilon_\theta^*$ eventually converges to the data score function:

$$\epsilon_\theta^*(\boldsymbol{x}_t|\boldsymbol{c}) = -\sigma_t \nabla_{\boldsymbol{x}_t} \log p_t(\boldsymbol{x}_t|\boldsymbol{c}) \qquad (2)$$

Given condition $\boldsymbol{c}$, $\epsilon_\theta$ can be leveraged to generate images from $p_\theta(\boldsymbol{x}|\boldsymbol{c})$ by denoising noises from $p_1$ iteratively.

**Discrete AR & masked models.** AR models (Chen et al., 2020) and masked-prediction models (Chang et al., 2022) function similarly. Both discretize images $\boldsymbol{x}$ into token sequences $\boldsymbol{x}_{1:N}$ and then perform token prediction. Their maximum likelihood training objective can be unified as

$$\min_\theta \mathbb{E}_{p(\boldsymbol{x}_{1:N}, \boldsymbol{c})} - \sum_i p_\theta(\boldsymbol{x}_n|\boldsymbol{x}_{<n}, \boldsymbol{c}). \qquad (3)$$

For AR models, $\boldsymbol{x}_{<n}$ represents the first $i$ tokens in a pre-determined order, and $\boldsymbol{x}_i$ is the next token to be predicted. For masked models, $\boldsymbol{x}_i$ represents all the unknown tokens that are randomized masked during training, while $\boldsymbol{x}_{<n}$ are the unmasked ones. Due to discrete modeling, the data likelihood $p_\theta$ in Eq. (3) can be easily calculated.

## 2.2. Classifier-Free Guidance

**Continuous CFG.** In diffusion modeling, vanilla temperature sampling (dividing model output by a constant value) is generally found ineffective in improving generation quality (Dhariwal & Nichol, 2021). Current methods typically employ CFG (Ho & Salimans, 2022), which redefines the sampling denoising function $\epsilon_\theta^s(\boldsymbol{x}_t|c)$ using two models:

$$\epsilon_\theta^s(\boldsymbol{x}_t|c) := \epsilon_\theta^c(\boldsymbol{x}_t|c) + s[\epsilon_\theta^c(\boldsymbol{x}_t|c) - \epsilon_\theta^u(\boldsymbol{x}_t)], \qquad (4)$$

where $\epsilon_\theta^c$ and $\epsilon_\theta^u$ respectively model the conditional data distribution $p(x|c)$ and the unconditional data distribution $p(x)$. In practice, $\epsilon_\theta^u$ can be jointly trained with $\epsilon_\theta^c$, by randomly masking the conditioning data in Eq. (1) with some fixed probability.

According to Eq. (2), CFG's sampling distribution $p^s(\boldsymbol{x}|c)$ has shifted from standard conditional distribution $p(\boldsymbol{x}|\boldsymbol{c})$ to

$$p^s(\boldsymbol{x}|c) \propto p(\boldsymbol{x}|\boldsymbol{c}) \left[ \frac{p(\boldsymbol{x}|\boldsymbol{c})}{p(\boldsymbol{x})} \right]^s. \qquad (5)$$

CFG offers an effective approach for lowering sampling temperature in visual generation by simply increasing $s > 0$, thereby substantially improving sample quality.

**Discrete CFG.** Besides diffusion, CFG is also a critical sampling technique in discrete visual modeling (Li et al., 2023; Team, 2024; Tian et al., 2024; Xie et al., 2024).

Though the guidance operation performs on the logit space instead of the score field:

$$\ell_\theta^s(\boldsymbol{x}_n|\boldsymbol{x}_{<n}, \boldsymbol{c}) = \\ \ell_\theta^c(\boldsymbol{x}_n|\boldsymbol{x}_{<n}, \boldsymbol{c}) + s[\ell_\theta^c(\boldsymbol{x}_n|\boldsymbol{x}_{<n}, \boldsymbol{c}) - \ell_\theta^u(\boldsymbol{x}_n|\boldsymbol{x}_{<n})].$$

Given $\ell_\theta \propto \log p_\theta$, it can be proved the sampling distribution for discrete visual models also satisfies Eq. (5).

## 3. Method

Despite its effectiveness, CFG requires inferencing an extra unconditional model to guide the sampling process, directly doubling the computation cost. Moreover, CFG complicates the post-training of visual generative models because the unconditional model needs to be additionally considered in algorithm design (Meng et al., 2023; Black et al., 2023).

We propose **Guidance-Free Training (GFT)** as an alternative method of CFG for improving sample quality in visual generation without guided sampling. GFT matches CFG in performance but only leverages a single model to represent CFG's sampling distribution $p^s(\boldsymbol{x}|c)$.

We derive GFT's training objective for diffusion models in Sec. 3.1, discuss its practical implementation in Sec. 3.2, and explain how it can be extended to discrete AR and masked models in Sec. 3.3.

## 3.1. Algorithm Derivation

The key challenge to directly learn the target sampling model $\epsilon_\theta^s$ is the absence of a dataset that aligns with the distribution $p^s(\boldsymbol{x}|c)$ in Eq. (5). This makes it impractical to optimize a maximum-likelihood-training objective like

$$\min_\theta \mathbb{E}_{p^s(\boldsymbol{x}, \boldsymbol{c}), t, \boldsymbol{\epsilon}} \left[ \|\epsilon_\theta^s(\boldsymbol{x}_t|c) - \boldsymbol{\epsilon}\|_2^2 \right],$$

as we cannot draw samples from $p^s$. In contrast, training $\epsilon_\theta^c(\boldsymbol{x}|\boldsymbol{c})$ and $\epsilon_\theta^u(\boldsymbol{x})$ separately as in CFG is feasible because their corresponding datasets, $\{(\boldsymbol{x}, \boldsymbol{c}) \sim p(\boldsymbol{x}, \boldsymbol{c})\}$ and $\{\boldsymbol{x} \sim p(\boldsymbol{x})\}$, can be easily obtained.

To address this, we reformulate Eq. (4):

$$\underbrace{\epsilon_\theta^c(\boldsymbol{x}_t|c)}_{\substack{p(\boldsymbol{x}|\boldsymbol{c}) \\ \text{Learnable}}} = \frac{1}{1+s} \underbrace{\epsilon_\theta^s(\boldsymbol{x}_t|c)}_{\substack{p^s(\boldsymbol{x}|\boldsymbol{c}) \\ \text{Target sampling model}}} + \frac{s}{1+s} \underbrace{\epsilon_\theta^u(\boldsymbol{x}_t)}_{\substack{p(\boldsymbol{x}) \\ \text{Learnable}}}. \qquad (6)$$

Although learning $\epsilon_\theta^s$ directly is difficult, we note it can be combined with an unconditional model $\epsilon_\theta^u$ to represent the standard conditional $\epsilon_\theta^c$, which is learnable. Thus, we can leverage the *same* conditional loss in Eq. (1) to train $\epsilon_\theta^s$, namely:

$$\min_\theta \mathbb{E}_{p(\boldsymbol{x}, \boldsymbol{c}), t, \boldsymbol{\epsilon}} \left[ \|\frac{1}{1+s}\epsilon_\theta^s(\boldsymbol{x}_t|c) + \frac{s}{1+s}\epsilon_\theta^u(\boldsymbol{x}_t) - \boldsymbol{\epsilon}\|_2^2 \right], \qquad (7)$$

**Algorithm 1** Guidance-Free Training (Diffusion)

---

1: Initialize $\theta$ from pretrained models or from scratch.
2: **for** each gradient step **do**
3:     *// CFG training w/ pseudo-temperature $\beta$*
4:     $\boldsymbol{x}, \boldsymbol{c} \sim p(\boldsymbol{x}, \boldsymbol{c})$
5:     $\beta \sim U(0,1), t \sim U(0,1)$
6:     $\boldsymbol{\epsilon} \sim \mathcal{N}(\boldsymbol{0}, \boldsymbol{I}^2)$
7:     $\boldsymbol{x}_t = \alpha_t \boldsymbol{x} + \sigma_t \boldsymbol{\epsilon}$
8:     $\boldsymbol{c}_\varnothing = \boldsymbol{c}$ masked by $\varnothing$ with 10% probability
9:     Calculate $\epsilon_\theta^s(\boldsymbol{x}_t | \boldsymbol{c}_\varnothing, \beta)$ in *training* mode
10:     *// Additional to CFG*
11:     Calculate $\epsilon_\theta^u(\boldsymbol{x}_t | \varnothing, 1)$ in *evaluation* mode
12:     $\epsilon_\theta = \beta \epsilon_\theta^s(\boldsymbol{x}_t | \boldsymbol{c}_\varnothing, \beta) + (1 - \beta)\mathbf{sg}[\epsilon_\theta^u(\boldsymbol{x}_t | \varnothing, 1)]$
13:     *// Standard Maximum Likelihood Training*
14:     $\theta \leftarrow \theta - \lambda \nabla_\theta \|\epsilon_\theta - \boldsymbol{\epsilon}\|_2^2$   (Eq. 9)
15: **end for**

---

where $\boldsymbol{\epsilon}$ is standard Gaussian noise, $\boldsymbol{x}_t = \alpha_t \boldsymbol{x} + \sigma_t \boldsymbol{\epsilon}$ are diffused images. $\alpha_t$ and $\sigma_t$ define the forward process.

To this end, we have a practical algorithm for directly learning guidance-free models $\epsilon_\theta^s$. However, unlike CFG which allows controlling sampling temperature by adjusting guidance scale $s$ to trade off fidelity and diversity, our method still lacks similar inference-time flexibility as Eq. (7) is performed for a specific $s$.

To solve this problem, we define a pseudo-temperature $\beta := 1/(1+s)$ and further condition our sampling model $\epsilon_\theta^s(\boldsymbol{x}_t | \boldsymbol{c}, \beta)$ on the extra $\beta$ input. We can randomly sample $\beta \in [0,1]$ during training, corresponding to $s \in [0, +\infty)$. The GFT objective in Eq. (7) now becomes:

$$\min_\theta \mathbb{E}_{p(\boldsymbol{x},\boldsymbol{c}),t,\boldsymbol{\epsilon},\beta} \left[ \|\beta \epsilon_\theta^s(\boldsymbol{x}_t | \boldsymbol{c}, \beta) + (1 - \beta)\epsilon_\theta^u(\boldsymbol{x}_t) - \boldsymbol{\epsilon}\|_2^2 \right]. \tag{8}$$

When $\beta = 1$, Eq. (8) reduces to conditional diffusion loss $\|\epsilon_\theta^s(\boldsymbol{x}_t | \boldsymbol{c}, \beta) - \boldsymbol{\epsilon}\|_2^2$. When $\beta = 0$, Eq. (8) becomes an unconditional loss $\|\epsilon_\theta^u(\boldsymbol{x}_t) - \boldsymbol{\epsilon}\|_2^2$. This allows simultaneous training of both conditional and unconditional models.

As pseudo-temperature $\beta$ decreases $1 \rightarrow 0$, the modeling target for $\epsilon_\theta^s$ gradually shifts from conditional data distribution $p(\boldsymbol{x}|\boldsymbol{c})$ to lower-temperature distribution $p^s(\boldsymbol{x}|\boldsymbol{c})$ as defined by Eq. (5) (See Fig. 2).

### 3.2. Practical Implementation

A desirable algorithm should not only ensure soundness but also offer computational efficiency, seamless integration, and practical deployability. To achieve this, we further present Eq. (9) as a practical loss function of GFT. The implementation is in Algorithm 1.

$$\mathcal{L}_\theta^{\text{diff}}(\boldsymbol{x}, \boldsymbol{c}_\varnothing, t, \boldsymbol{\epsilon}, \beta)$$
$$= \|\beta \epsilon_\theta^s(\boldsymbol{x}_t | \boldsymbol{c}_\varnothing, \beta) + (1 - \beta)\mathbf{sg}[\epsilon_\theta^u(\boldsymbol{x}_t | \varnothing, 1)] - \boldsymbol{\epsilon}\|_2^2. \tag{9}$$

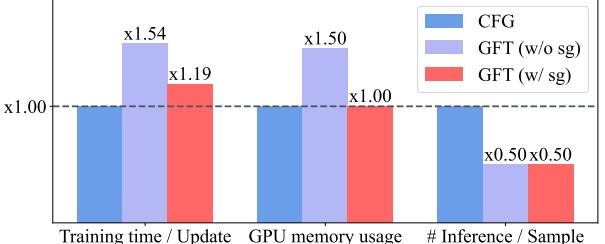

Figure 3: Comparison of computational efficiency between GFT and CFG. Estimated based on the DiT-XL model.

**Stopping the unconditional gradient.** The main difference between Eq. (9) and Eq. (8) is that $\epsilon_\theta^u$ is computed in evaluation mode, with model gradients stopped by the $\mathbf{sg}[\cdot]$ operation. To train the model unconditionally, we randomly mask conditions $\boldsymbol{c}$ with $\varnothing$ when computing $\epsilon_\theta^s$. We show this design does not affect the training convergence point:

**Theorem 1** (GFT Optimal Solution). *Given unlimited model capacity and training data, the optimal $\epsilon_{\theta*}^s$ for optimizing Eq. (9) and Eq. (8) are the same. Both satisfy*

$$\epsilon_{\theta*}^s(\boldsymbol{x}_t | \boldsymbol{c}, \beta)$$
$$= -\sigma_t \left[ \frac{1}{\beta} \nabla_{\boldsymbol{x}_t} \log p_t(\boldsymbol{x}_t | \boldsymbol{c}) - (\frac{1}{\beta} - 1) \nabla_{\boldsymbol{x}_t} \log p_t(\boldsymbol{x}_t) \right]$$

*Proof.* In Appendix B. $\square$

The stopping-gradient technique has the following benefits:

*(1) Alignment with CFG.* The practical GFT algorithm (Eq. 9) differs from CFG training by a single unconditional inference step. This allows us to implement GFT with only a few lines of code based on existing codebases.

*(2) Computational efficiency.* Since the extra unconditional calculation is gradient-free. GFT requires virtually no extra GPU memory and only 19% additional train time per update vs. CFG (Figure 3). This stands in contrast with the naive implementation without gradient stopping (Eq. 8), which is equivalent to doubling the batch size for CFG training.

*(3) Training stability.* We empirically observe that stopping the gradient for the unconditional model could lead to better training stability and thus improved performance.

**Input of $\beta$.** GFT requires an extra pseudo-temperature input in comparison with CFG. For this, we first process $\beta$ using the similar Fourier embedding method for diffusion time $t$ (Dhariwal & Nichol, 2021; Meng et al., 2023). This is followed by some MLP layers. Finally, the temperature embedding is added to the model's original time or class embedding. If fine-tuning, we apply zero initialization for the final MLP layer so that $\beta$ would not affect model output at the start of training.

Table 1: Comparison of GFT (ours) and other guidance-free methods. Numbers are reported based on experiments for the DiT-XL model or the VAR-d30 model. We use $8\times80$GB H100 GPU cards.

| Method | Guidance Distillation | Condition Contrastive Alignment | Guidance-Free Training |
|---|---|---|---|
| Modeling target | $p_\phi^c + p_\phi^u \rightarrow p_\theta^s(\boldsymbol{x}|\boldsymbol{c})$ | $p_\theta^s + p_\phi^c \rightarrow \frac{p(\boldsymbol{x}|\boldsymbol{c})}{p(\boldsymbol{x})}$ | $p_\theta^s + p_\theta^u \rightarrow p(\boldsymbol{x}|\boldsymbol{c})$ |
| Applicable area | Diffusion | AR & Masked | All |
| Loss form | $\left\|\boldsymbol{\epsilon}_\theta^s - [(1+s)\boldsymbol{\epsilon}_\phi^c - s\boldsymbol{\epsilon}_\phi^u]\right\|_2^2$ | $-\log\sigma(r_\theta^p)-\log\sigma(-r_\theta^n); r_\theta = \frac{1}{s}\log\frac{p_\theta^s}{p_\phi^c}$ | $\beta f_\theta^s + (1-\beta)f_\theta^u$ + Diff/AR Loss |
| Reference equation | Eq. (13) | Eq. (14) | Eq. (9) & Eq. (11) |
| Train from scratch? | Not allowed | Not allowed | Allowed |
| # Inference / Update | 3 | 4 | 2 |
| Train time / Update | $\times1.19$ | $\times1.69$ | $\times1.00$ |
| GPU memory usage | $\times1.15$ | $\times1.39$ | $\times1.00$ |

**Hyperparameters.** Due to the high similarity between CFG and GFT training, we inherit most hyperparameter choices used for existing CFG models and mainly adjust parameters like learning rate during finetuning. When training from scratch, we find simply keeping all parameters the same with CFG is enough to yield good performance.

**Training epochs.** When fine-tuning pretrained CFG models, we find 1% - 5% of pretraining epochs are sufficient to achieve nearly lossless FID performance. When from scratch, we always use the same training epochs compared with the CFG baseline.

### 3.3. GFT for AR and Masked Models

Similar to Sec. 3.1, we can derive the GFT objective for AR and masked models as standard cross-entropy loss:

$$\mathcal{L}_\theta^{\mathrm{AR}}(\boldsymbol{x}, \boldsymbol{c}_\varnothing, \beta) = -\sum_i \log p_\theta^c(\boldsymbol{x}_n|\boldsymbol{x}_{<n}, \boldsymbol{c}_\varnothing, \beta) \quad (10)$$

$$= -\sum_i \log \frac{e^{\ell_\theta^c(\boldsymbol{x}_n|\boldsymbol{x}_{<n}, \boldsymbol{c}_\varnothing, \beta)}}{\sum_{w\in\mathcal{V}} e^{\ell_\theta^c(w|\boldsymbol{x}_{<n}, \boldsymbol{c}_\varnothing, \beta)}}, \quad (11)$$

where $w$ is a token in the vocabulary $\mathcal{V}$, and

$$\ell_\theta^c(w|\boldsymbol{x}_{<n}, \boldsymbol{c}_\varnothing, \beta)$$
$$:= \beta\ell_\theta^s(w|\boldsymbol{x}_{<n}, \boldsymbol{c}_\varnothing, \beta) + (1-\beta)\mathrm{sg}[\ell_\theta^u(w|\boldsymbol{x}_{<n})]. \quad (12)$$

In Sec. 5, we apply GFT to a wide spectrum of visual generative models, including diffusion, AR, and masked models, demonstrating its versatility.

## 4. Connection with Other Guidance-Free Methods

Previous attempts to remove guided sampling from visual generation mainly include distillation methods for diffusion

models and alignment methods for AR models. Alongside GFT, these methods all transform the sampling distribution $p^s$ into simpler, learnable forms, differing mainly in how they decompose the sampling distribution and set up modeling targets (Table 1).

**Guidance Distillation** (Meng et al., 2023) is quite straightforward, it simply learns a single model to match the output of pretrained CFG targets using L2 loss:

$$\mathcal{L}_\theta^{\mathrm{GD}} = \left\|\boldsymbol{\epsilon}_\theta^s(\boldsymbol{x}_t|\boldsymbol{c}, s) - [(1+s)\boldsymbol{\epsilon}_\phi^c(\boldsymbol{x}_t|\boldsymbol{c}) - s\boldsymbol{\epsilon}_\phi^u(\boldsymbol{x}_t)]\right\|_2^2, \quad (13)$$

where $\boldsymbol{\epsilon}_\phi^u$ and $\boldsymbol{\epsilon}_\phi^c$ are pretrained models. $\mathcal{L}_\theta^{\mathrm{GD}}$ breaks down the sampling model into a linear combination of conditional and unconditional models, which can be separately learned.

Despite being effective, Guidance distillation relies on pretrained CFG models as teacher models, and cannot be leveraged for from-scratch training. This results in an indirect, two-stage pipeline for learning guidance-free models. In comparison, our method unifies guidance-free training in one singular loss, allowing learning in an end-to-end style. Besides, GFT no longer requires learning an explicit conditional model $\boldsymbol{\epsilon}_\theta^c$. This saves training computation and VRAM usage. A detailed comparison is in Table 1.

**Condition Contrastive Alignment** (Chen et al., 2024b) constructs a preference pair for each image $\boldsymbol{x}$ in the dataset and applies similar preference alignment techniques for language models (Rafailov et al., 2023; Chen et al., 2024a) to fine-tune visual AR models:

$$\mathcal{L}_\theta^{\mathrm{CCA}} = -\log\sigma\left[r_\theta(\boldsymbol{x}, \boldsymbol{c}^p)\right] - \log\sigma\left[-r_\theta(\boldsymbol{x}, \boldsymbol{c}^n)\right], \quad (14)$$

where $\boldsymbol{c}^p$ is the preferred positive condition corresponding to the image $\boldsymbol{x}$, $\boldsymbol{c}^n$ is a negative condition randomly and independently sampled from the dataset. Given a conditional

Table 2: Model comparisons on the class-conditional ImageNet $256 \times 256$ benchmark.

| Model Type | FID ↓ | |
| --- | --- | --- |
| | w/o Guidance | w/ Guidance |
| **Diffusion Models** | | |
| ADM (Dhariwal & Nichol, 2021) | 7.49 | 3.94 |
| LDM-4 (Rombach et al., 2022) | 10.56 | 3.60 |
| U-ViT-H/2 (Bao et al., 2023) | – | 2.29 |
| MDTv2-XL/2 (Gao et al., 2023) | 5.06 | 1.58 |
| DiT-XL/2 (Peebles & Xie, 2023) | 9.34 | 2.11 |
| +Distillation (Meng et al., 2023) | 2.11 | – |
| **+GFT (Ours)** | **1.99** | – |
| **Autoregressive Models** | | |
| VQGAN (Esser et al., 2021) | 15.78 | 5.20 |
| ViT-VQGAN (Yu et al., 2021) | 4.17 | 3.04 |
| RQ-Transformer (Lee et al., 2022) | 7.55 | 3.80 |
| LlamaGen-3B (Sun et al., 2024) | 9.44 | 2.22 |
| +CCA (Chen et al., 2024b) | 2.69 | – |
| **+GFT (Ours)** | 2.21 | – |
| VAR-d30 (Tian et al., 2024) | 5.26 | 1.92 |
| +CCA (Chen et al., 2024b) | 2.54 | – |
| **+GFT (Ours)** | **1.91** | – |
| **Masked Models** | | |
| MaskGIT (Chang et al., 2022) | 6.18 | – |
| MAGVIT-v2 (Yu et al., 2023b) | 3.65 | 1.78 |
| MAGE (Li et al., 2023) | 6.93 | – |
| MAR-B (Li et al., 2024) | 4.17 | 2.27 |
| **+GFT (Ours)** | **2.39** | – |

reference model $p_\phi^c$, the implicit reward $r_\theta$ is defined as

$$r_\theta(\boldsymbol{x}, \boldsymbol{c}) := \frac{1}{s} \log \frac{p_\theta^s(\boldsymbol{x}|\boldsymbol{c})}{p_\phi^c(\boldsymbol{x}|\boldsymbol{c})}.$$

CCA proves the optimal solution for solving Eq. (14) is $r_\theta^* = \log \frac{p(\boldsymbol{x}|\boldsymbol{c})}{p(\boldsymbol{x})}$, thus the convergence point for $p_\theta^s(\boldsymbol{x}|\boldsymbol{c})$ also satisfies Eq. (5).

Both CCA and GFT train sampling model $p_\theta^s$ directly by combining it with another model to represent a learnable distribution. GFT leverages $\beta p_\theta^s + (1 - \beta)p_\theta^u$ to represent standard conditional distribution $p(\boldsymbol{x}|\boldsymbol{c})$, while CCA combines $p_\theta^s$ and pretrained $p_\phi(\boldsymbol{x}|\boldsymbol{c})$ to represent the conditional residual $\log \frac{p(\boldsymbol{x}|\boldsymbol{c})}{p(\boldsymbol{x})}$. They also differ in applicable areas. CCA is based on language alignment losses, which requires calculating model likelihood $\log p_\theta$ during training. This forbids its direct application to diffusion models, where calculating exact likelihood is infeasible.

# 5. Experiments

Our experiments aim to investigate:

1. GFT's effectiveness and efficiency in fine-tuning CFG models into guidance-free variants (Sec. 5.2)

2. GFT's ability in training guidance-free models from scratch, compared with classic CFG training (Sec. 5.3)

Table 3: Model comparisons for zero-shot text-to-image generation on the COCO 2014 validation set.

| Text to Image Models | FID ↓ | |
| --- | --- | --- |
| | w/o Guidance | w/ Guidance |
| GLIDE (Nichol et al., 2021) | – | 12.24 |
| LDM (Rombach et al., 2022) | – | 12.63 |
| DALL·E 2 (Ramesh et al., 2022) | – | 10.39 |
| Stable Diffusion 1.5 (Rombach et al., 2022) | 22.55 | 7.87 |
| +Distillation (Meng et al., 2023) | 8.16 | – |
| **+GFT (Ours)** | **8.10** | – |

3. GFT's capability of controlling diversity-fidelity trade-off through temperature parameter $\beta$. (Sec. 5.4)

## 5.1. Experimental Setups

**Tasks & Models.** We evaluate GFT in both class-to-image (C2I) and text-to-image (T2I) tasks. For C2I, we experiment with diverse architectures: DiT (Peebles & Xie, 2023) (transformer-based latent diffusion model), MAR (Li et al., 2024) (masked-token prediction model with diffusion heads), and autoregressive models: VAR (Tian et al., 2024) and LlamaGen (Sun et al., 2024). For T2I, we use Stable Diffusion 1.5 (Rombach et al., 2022), a text-to-image model based on the U-Net architecture (Ronneberger et al., 2015), to provide a comprehensive evaluation of GFT's performance across various conditioning modalities. All these models rely on guided sampling as a critical component.

**Training & Evaluation.** We train C2I models on ImageNet-256x256 (Deng et al., 2009). For T2I models, we use a subset of the LAION-Aesthetic 5+ (Schuhmann et al., 2022), consisting of 18 million image-text pairs. Our codebases are directly modified from the official CFG implementation of each respective baseline, keeping most hyperparameters consistent with CFG training. We use official OPENAI evaluation scripts to evaluate our C2I models. For T2I models, we evaluate our model on zero-shot COCO 2014 (Lin et al., 2014). The training and evaluation details for each model can be found in Appendix D.

## 5.2. Make CFG Models Guidance-Free

**Method Effectiveness.** In Table 2 and 3, we apply GFT to fine-tune a wide spectrum of visual generative models. With less than 5% pretraining computation, the fine-tuned models achieve comparable FID scores to CFG while being 2× faster in sampling. Figure 4 visually demonstrates this quality improvement.

**Comparison with other guidance-free approaches.** GFT achieves comparable performance to guidance distillation (designed for diffusion models) and outperforms AR alignment method (Table 2 and 3). Notably, GFT demon-

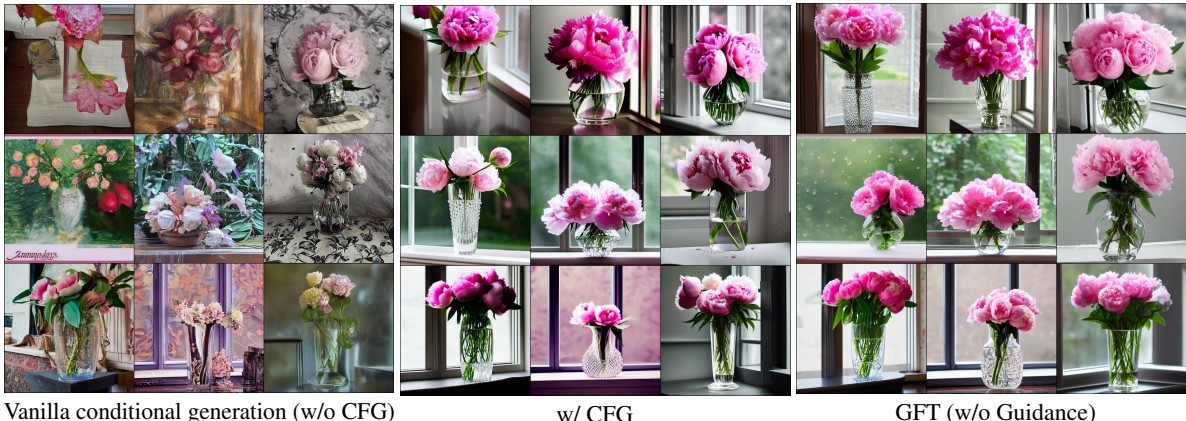

| Vanilla conditional generation (w/o CFG) | w/ CFG | GFT (w/o Guidance) |

Figure 4: Qualitative T2I comparison between vanilla conditional generation, GFT, and CFG on Stable Diffusion 1.5 with the prompt "Elegant crystal vase holding pink peonies, soft raindrops tracing paths down the window behind it". More examples are in Figure 14.

strates superior efficiency compared to both methods (Table 1). We attribute this effectiveness to GFT's end-to-end training style, and to the nice convergence property of its maximum likelihood training objective.

**Finetuning Efficiency** Figure 5 tracks the FID progression of DiT-XL/2 during fine-tuning. The guidance-free FID rapidly improves from 9.34 to 2.22 in the first epoch, followed by steady optimization. After three epochs, our model achieves a better FID than the CFG baseline (2.05 vs 2.11). This computational is almost negligible compared with pretraining, demonstrating GFT's efficiency.

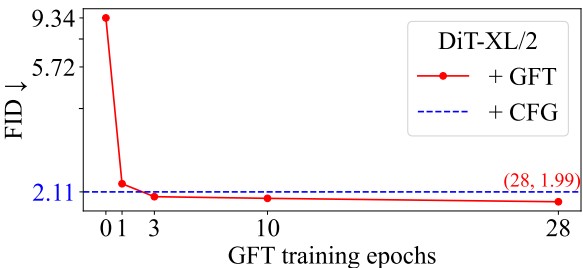

Figure 5: Efficient convergence of FID scores for GFT using DiT-XL/2 model across training epochs.

### 5.3. Building Guidance-Free Models from Scratch

Training Guidance-Free Models from scratch is more tempting than the two-stage pipeline adopted by Sec 5.2. However, this is also more challenging due to higher requirements for the algorithm's stability and convergence speed. We investigate this by comparing from-scratch GFT training with classic supervised training using CFG across various architectures, maintaining consistent training epochs. We mainly focus on smaller models due to computational constraints.

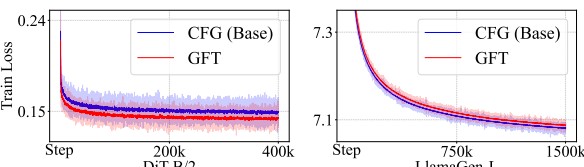

Figure 6: Comparison of convergence speed between GFT and CFG on diffusion and autoregressive models in from-scratch training.

**Performance.** Table 4 shows that GFT models trained from scratch outperform CFG baselines across DiT-B/2, MAR-B, and LlamaGen-L models, while reducing evaluation costs by 50%. Notably, these from-scratch models outperform their fine-tuned counterparts, demonstrating the advantages of direct guidance-free training.

**Training stability.** An informative indicator of an algorithm's stability and scalability is its loss convergence speed. With consistent hyperparameters, we find GFT convergences at least as fast as CFG for both diffusion and autoregressive modeling (Figure 6). Direct loss comparison is valid as both methods optimize the same objective: the conditional modeling loss for the dataset distribution. The only difference is that the conditional model for CFG is a single end-to-end model, while for GFT it is constructed as a linear interpolation of two model outputs.

Based on the above observations, we believe that GFT is at least as stable and reliable as CFG algorithms, providing a new training paradigm and a viable alternative for visual generative models.

| Method | Guidance | DiT-B/2 (Peebles & Xie, 2023) | | MAR-B (Li et al., 2024) | | LlamaGen-L (Sun et al., 2024) | | VAR-d16 (Tian et al., 2024) | |
|---|---|---|---|---|---|---|---|---|---|
| | | FID ↓ | IS ↑ | FID ↓ | IS ↑ | FID ↓ | IS ↑ | FID ↓ | IS ↑ |
| Base | w/o | 44.8 | 30.7 | 4.17 | 175.4 | 19.0 | 64.7 | 11.97 | 105.5 |
| +CFG | w/ | 9.72 | 161.5 | **2.27** | 260.7 | 3.06 | 257.1 | 3.36 | **280.1** |
| +GFT | w/o | 10.9 | 128.5 | 2.39 | 264.7 | 2.88 | 238.4 | **3.28** | 251.3 |
| GFT | w/o | **9.04** | **166.6** | **2.27** | **279.5** | **2.52** | **270.5** | 3.42 | 254.8 |

Table 4: Performance comparison between GFT from-scratch training, CFG, and GFT fine-tuning variants across different model architectures. GFT and the base model are trained for the same number of epochs.

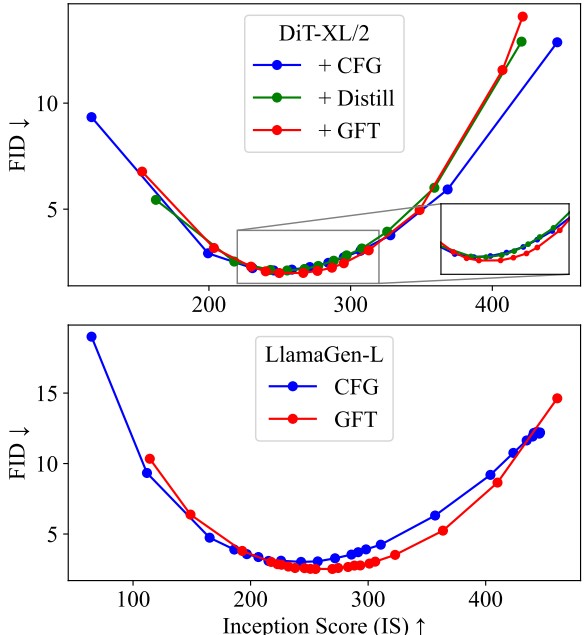

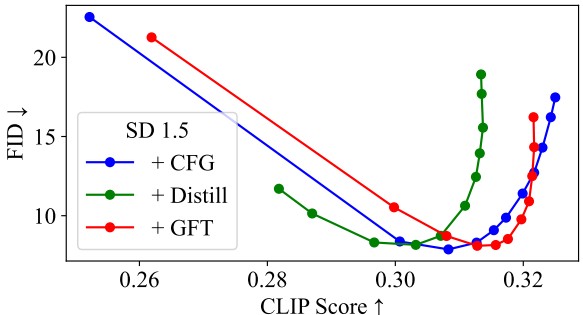

Figure 7: FID-IS trade-off comparisons on ImageNet. Upper: DiT-XL/2 with GFT (fine-tuned), CFG, and Guidance Distillation. Lower: LlamaGen-L with GFT (trained from scratch) and CFG.

Figure 8: FID-CLIP trade-off comparison on COCO-2014 validation set. Methods compared using Stable Diffusion 1.5.

## 5.4. Sampling Temperature for Visual Generation

A key advantage of CFG is its flexible sampling temperature for diversity-fidelity trade-offs. Our results demonstrate that GFT models share this capability.

We evaluate diversity-fidelity trade-offs across various models, with FID-IS trade-off for c2i models and FID-CLIP trade-off for t2i models. Results for DiT-XL/2 (fine-tuning), LlamaGen-L (from-scratch training) and Stable Diffusion 1.5 (fine-tuning) are shown in Figures 7 and 8, with additional trade-off curves provided in Appendix C. For DiT-XL/2 and Stable Diffusion 1.5, we also compare GFT with Guidance Distillation, showing that GFT achieves results comparable to CFG while outperforming Guidance Distillation on CLIP scores.

Figure 2 shows how adjusting temperature $\beta$ produces effects similar to adjusting CFG's scale $s$. This similarity results from both methods aiming to model the same distribution (Eq. 5). The key difference is that GFT directly learns a series of sampling distributions controlled by $\beta$ through training, while CFG modifies the sampling process to achieve comparable results.

## 6. Conclusion

In this work, we proposed Guidance-Free Training (GFT) as an alternative to guided sampling in visual generative models, achieving comparable performance to Classifier-Free Guidance (CFG). GFT reduces sampling computational costs by 50%. The method is simple to implement, requiring minimal modifications to existing codebases. Unlike previous distillation-based methods, GFT enables direct training from scratch.

Our extensive evaluation across multiple types of visual models demonstrates GFT's effectiveness. The approach maintains high sample quality while offering flexible control over the diversity-fidelity trade-off through temperature adjustment. GFT represents an advancement in making high-quality visual generation more efficient and accessible.

## Acknowledgement

We thank Huanran Chen, Xiaoshi Wu, Cheng Lu, Fan Bao, Chengdong Xiang, Zhengyi Wang, Chang Li and Peize Sun for the discussion. This work was supported by the NSFC Project (No. 62376131, 92270001, 92370124, 92248303) and the High Performance Computing Center, Tsinghua University. J.Z is also supported by the XPlorer Prize.

## Impact Statement

Our Guidance-Free Training (GFT) method significantly reduces the computational costs of visual generative models by eliminating the need for dual inference during sampling, contributing to more sustainable AI development and reduced environmental impact. However, since our method accelerates the sampling process of generative models, it could potentially be misused to create harmful content more efficiently, emphasizing the importance of establishing appropriate safety measures and deploying these models responsibly with proper oversight mechanisms.

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

# A. Related Work

**Visual generation model with guidance.**    Visual generative modeling has witnessed significant advancements in recent years. Recent explicit-likelihood approaches can be broadly categorized into diffusion-based models (Sohl-Dickstein et al., 2015; Song & Ermon, 2019; Ho et al., 2020; Song et al., 2020b; Dhariwal & Nichol, 2021; Kingma et al., 2021; Rombach et al., 2022; Ramesh et al., 2022; Saharia et al., 2022; Karras et al., 2022; Bao et al., 2023; Peebles & Xie, 2023; Esser et al., 2024; Xie et al., 2024), auto-regressive models (Chen et al., 2020; Esser et al., 2021; Ramesh et al., 2021; Yu et al., 2021; Tian et al., 2024; Team, 2024; Sun et al., 2024; Ma et al., 2024; Zhang et al., 2024; Tang et al., 2024), and masked-prediction models (Chang et al., 2022; Yu et al., 2023a; Chang et al., 2023; Li et al., 2024; Yu et al., 2024; Fan et al., 2024). The introduction of guidance techniques has substantially improved the capabilities of these models. These include classifier guidance (Dhariwal & Nichol, 2021), classifier-free guidance (Ho & Salimans, 2022), energy guidance (Chung et al., 2022; Zhao et al., 2022; Lu et al., 2023; Song et al., 2023a), and various advanced guidance methods (Kim et al., 2022; Hong et al., 2023; Kynkäänniemi et al., 2024; Karras et al., 2024; Chung et al., 2024; Koulischer et al.; Shenoy et al., 2024).

**Guidance distillation.**    To address the computational overhead introduced by classifier-free guidance (CFG), One widely used approach to remove CFG is guidance distillation (Meng et al., 2023), where a student model is trained to directly learn the output of a pre-trained teacher model that incorporates guidance. This idea of guidance distillation has been widely adopted in methods aimed at accelerating diffusion models (Luo et al., 2023; Yin et al., 2024; Lin et al., 2024; Zhou et al., 2024a). By integrating the teacher model's guided outputs into the training process, these approaches achieve efficient few-step generation without guidance.

**Alternative Methods for Building Guidance-free Models.**    Recent studies in diffusion models show that perceptual losses (Lin & Yang, 2023), score-based distillation (Sauer et al., 2025a; Zhou et al., 2024c; Sauer et al., 2025b; Zhou et al., 2024b), and consistency models (Song et al., 2023b; Geng et al., 2024; Lu & Song, 2024) can also achieve comparable FID scores to CFG. For auto-regressive models, Condition Contrastive Alignment (Chen et al., 2024b) could enhance guidance-free performance through alignment (Rafailov et al., 2023; Chen et al., 2024a) in a self-contrastive manner.

# B. Proof of Theorem 1

We first copy the training objective in Eq. (9) and Eq. (8).

$$\mathcal{L}_\theta^{\mathrm{raw}} = \mathbb{E}_{p(\boldsymbol{x},\boldsymbol{c}),t,\boldsymbol{\epsilon},\beta} \left[ \|\beta\epsilon_\theta^s(\boldsymbol{x}_t|\boldsymbol{c},\beta) + (1-\beta)\epsilon_\theta^u(\boldsymbol{x}_t) - \boldsymbol{\epsilon}\|_2^2 \right]. \tag{15}$$

$$\mathcal{L}_\theta^{\mathrm{practical}} = \mathbb{E}_{p(\boldsymbol{x},\boldsymbol{c}_\varnothing),t,\boldsymbol{\epsilon},\beta} \|\beta\epsilon_\theta^s(\boldsymbol{x}_t|\boldsymbol{c}_\varnothing,\beta) + (1-\beta)\mathbf{sg}[\epsilon_\theta^u(\boldsymbol{x}_t|\varnothing,1)] - \boldsymbol{\epsilon}\|_2^2. \tag{16}$$

**Theorem 1** (GFT Optimal Solution). *Given unlimited model capacity and training data, the optimal $\epsilon_{\theta*}^s$ for optimizing Eq. (15) and Eq. (16) are the same. Both satisfy*

$$\epsilon_{\theta*}^s(\boldsymbol{x}_t|\boldsymbol{c},\beta) = -\sigma_t \left[ \frac{1}{\beta}\nabla_{\boldsymbol{x}_t}\log p_t(\boldsymbol{x}_t|\boldsymbol{c}) - (\frac{1}{\beta}-1)\nabla_{\boldsymbol{x}_t}\log p_t(\boldsymbol{x}_t) \right]$$

*Proof.* The proof is quite straightforward.

First consider the unconditional part of the model. Let $\beta = 1$ in $\mathcal{L}_\theta^{\mathrm{raw}}$, we have

$$\mathcal{L}_\theta^{\mathrm{raw}} = \mathbb{E}_{p(\boldsymbol{x},\boldsymbol{c}),t,\boldsymbol{\epsilon},\beta=1} \left[ \|\epsilon_\theta^u(\boldsymbol{x}_t) - \boldsymbol{\epsilon}\|_2^2 \right],$$

which is standard unconditional diffusion loss. According to Eq. (2) we have

$$\epsilon_{\theta*}^u(\boldsymbol{x}_t) = -\sigma_t \nabla_{\boldsymbol{x}_t}\log p_t(\boldsymbol{x}_t) \tag{17}$$

Then we prove for $\forall \beta \in (0,1]$, stopping the unconditional gradient does not change this optimal solution. Taking derivatives of $\mathcal{L}_\theta^{\mathrm{practical}}$ we have:

$$\begin{aligned}
\nabla_\theta \mathcal{L}_\theta^{\mathrm{practical}}(\boldsymbol{c}_\varnothing = \varnothing) &= \mathbb{E}_{p(\boldsymbol{x}),t,\boldsymbol{\epsilon},\beta}\nabla_\theta\|\beta\epsilon_\theta^s(\boldsymbol{x}_t|\varnothing,1) + (1-\beta)\mathbf{sg}[\epsilon_\theta^u(\boldsymbol{x}_t|\varnothing,1)] - \boldsymbol{\epsilon}\|_2^2 \\
&= \mathbb{E}_{p(\boldsymbol{x}),t,\boldsymbol{\epsilon},\beta}2\beta[\nabla_\theta\epsilon_\theta^s(\boldsymbol{x}_t|\varnothing,1)]\|\beta\epsilon_\theta^s(\boldsymbol{x}_t|\varnothing,1) + (1-\beta)\epsilon_\theta^u(\boldsymbol{x}_t|\varnothing,1) - \boldsymbol{\epsilon}\|_2 \\
&= \mathbb{E}_{p(\boldsymbol{x}),t,\boldsymbol{\epsilon},\beta}2\beta[\nabla_\theta\epsilon_\theta^s(\boldsymbol{x}_t|\varnothing,1)]\|\epsilon_\theta^s(\boldsymbol{x}_t|\varnothing,1) - \boldsymbol{\epsilon}\|_2 \\
&= [2\mathbb{E}\beta]\nabla_\theta\mathbb{E}_{p(\boldsymbol{x},\boldsymbol{c}),t,\boldsymbol{\epsilon},\beta=1}\left[\|\epsilon_\theta^u(\boldsymbol{x}_t) - \boldsymbol{\epsilon}\|_2^2\right] \\
&= [2\mathbb{E}\beta]\nabla_\theta\mathcal{L}_\theta^{\mathrm{raw}}(\beta=1)
\end{aligned}$$

Since $[2\mathbb{E}\beta]$ is a constant, this does not change the convergence point of $\mathcal{L}_\theta^{\mathrm{raw}}$. The optimal unconditional solution for $\mathcal{L}_\theta^{\mathrm{practical}}$ remains the same.

For the conditional part of the model, since both $\mathcal{L}_\theta^{\mathrm{raw}}$ and $\mathcal{L}_\theta^{\mathrm{practical}}$ are standard conditional diffusion loss, we have

$$\beta\epsilon_{\theta*}^s(\boldsymbol{x}_t|\boldsymbol{c},\beta) + (1-\beta)\epsilon_{\theta*}^u(\boldsymbol{x}_t) = -\sigma_t\nabla_{\boldsymbol{x}_t}\log p_t(\boldsymbol{x}_t|\boldsymbol{c})$$

Combining Eq. (17), we have

$$\epsilon_{\theta*}^s(\boldsymbol{x}_t|\boldsymbol{c},\beta) = -\sigma_t \left[ \frac{1}{\beta}\nabla_{\boldsymbol{x}_t}\log p_t(\boldsymbol{x}_t|\boldsymbol{c}) - (\frac{1}{\beta}-1)\nabla_{\boldsymbol{x}_t}\log p_t(\boldsymbol{x}_t) \right].$$

Let $s = \frac{1}{\beta} - 1$, we can see that GFT models the same sampling distribution as CFG (Eq. 4). $\qquad \square$

## C. Additional Experiment Results.

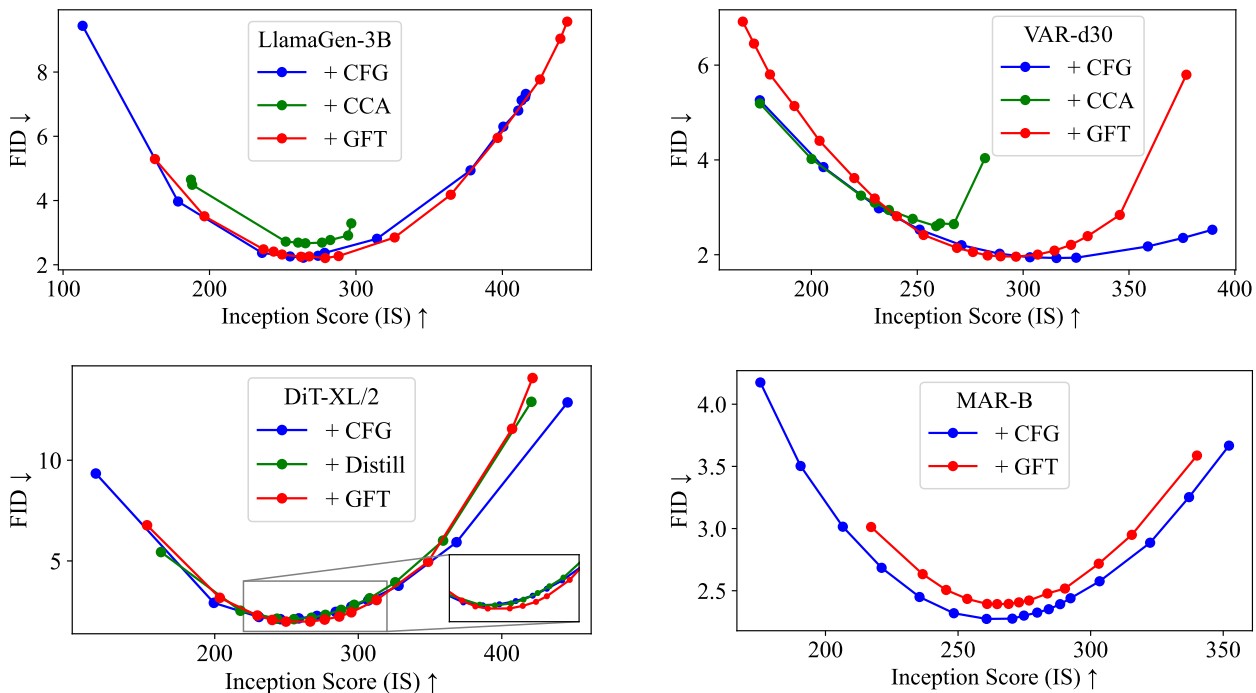

Figure 9: FID-IS trade-off comparison in fine-tuning experiments.

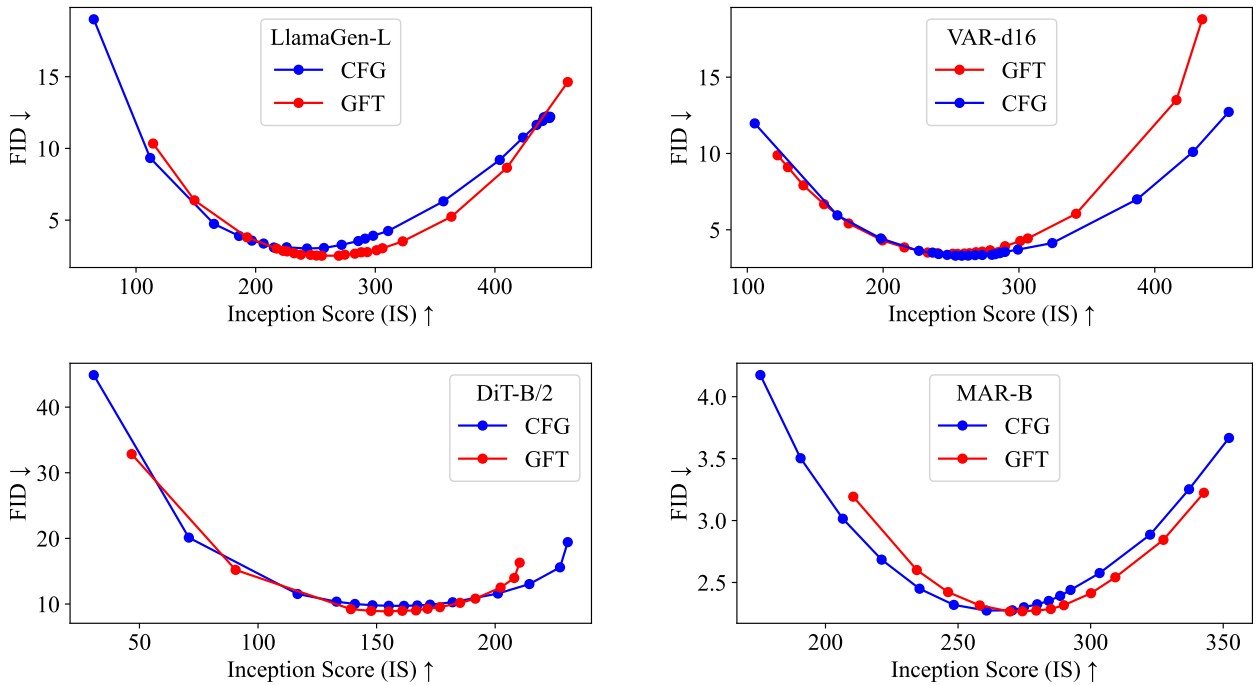

Figure 10: FID-IS trade-off comparison in from-scratch-training experiments.

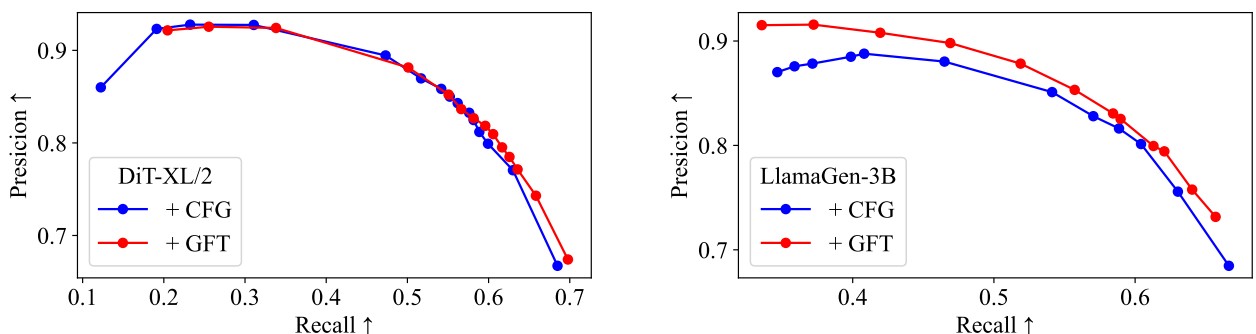

Figure 11: Precision-Recall trade-off comparison for DiT and LlamaGen in fine-tuning experiments.

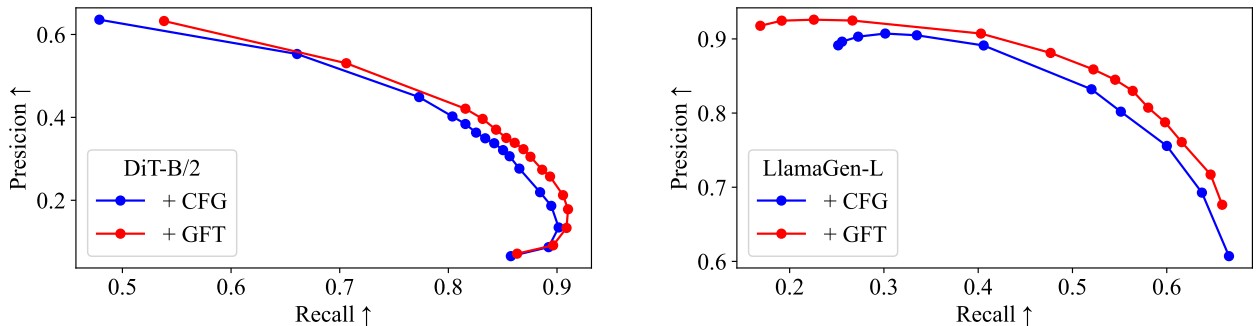

Figure 12: Precision-Recall trade-off comparison for DiT and LlamaGen in from-scratch-training experiments.

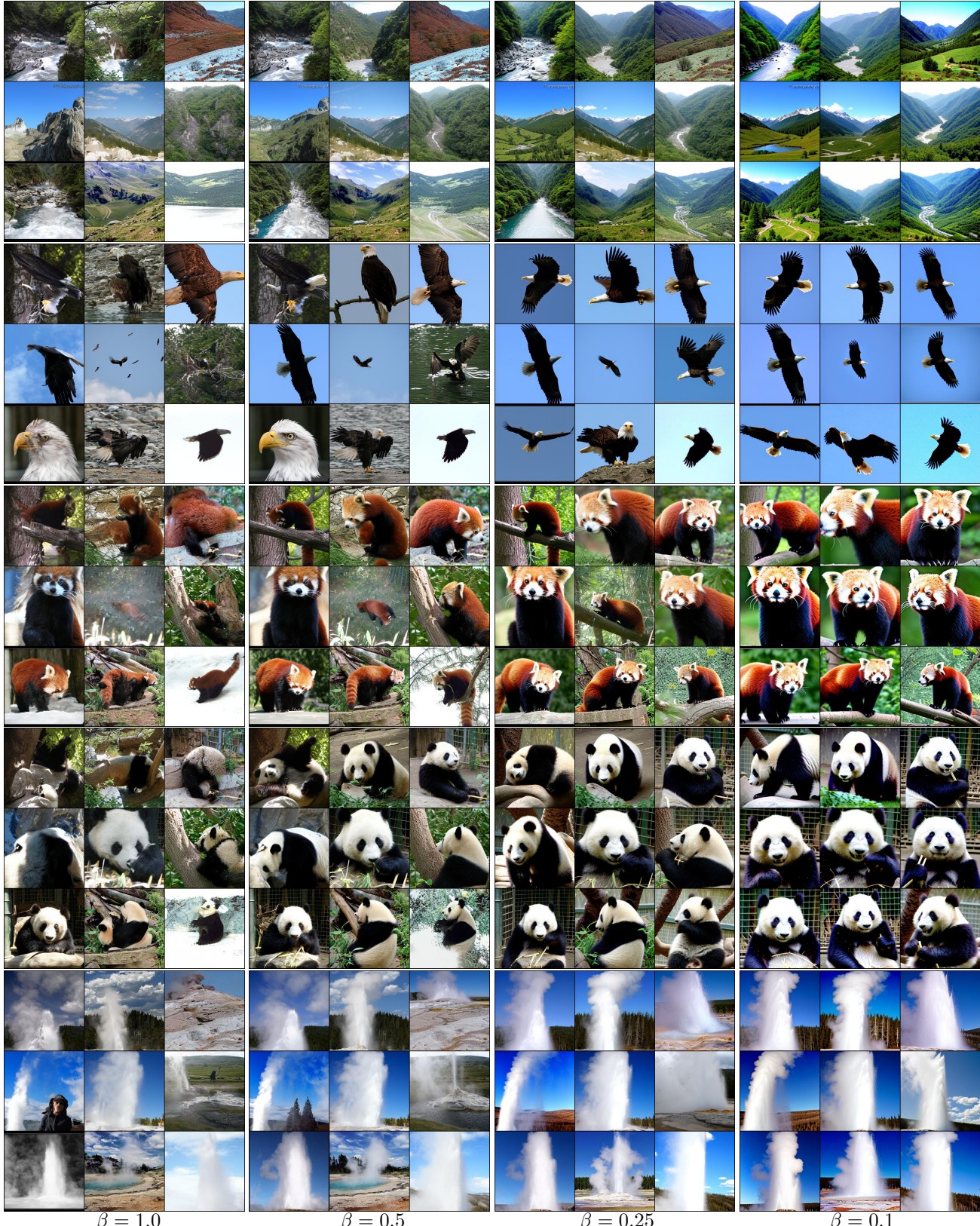

Figure 13: Additional results of temperature sampling ($\beta$) impact on DiT-XL/2 after applying GFT.

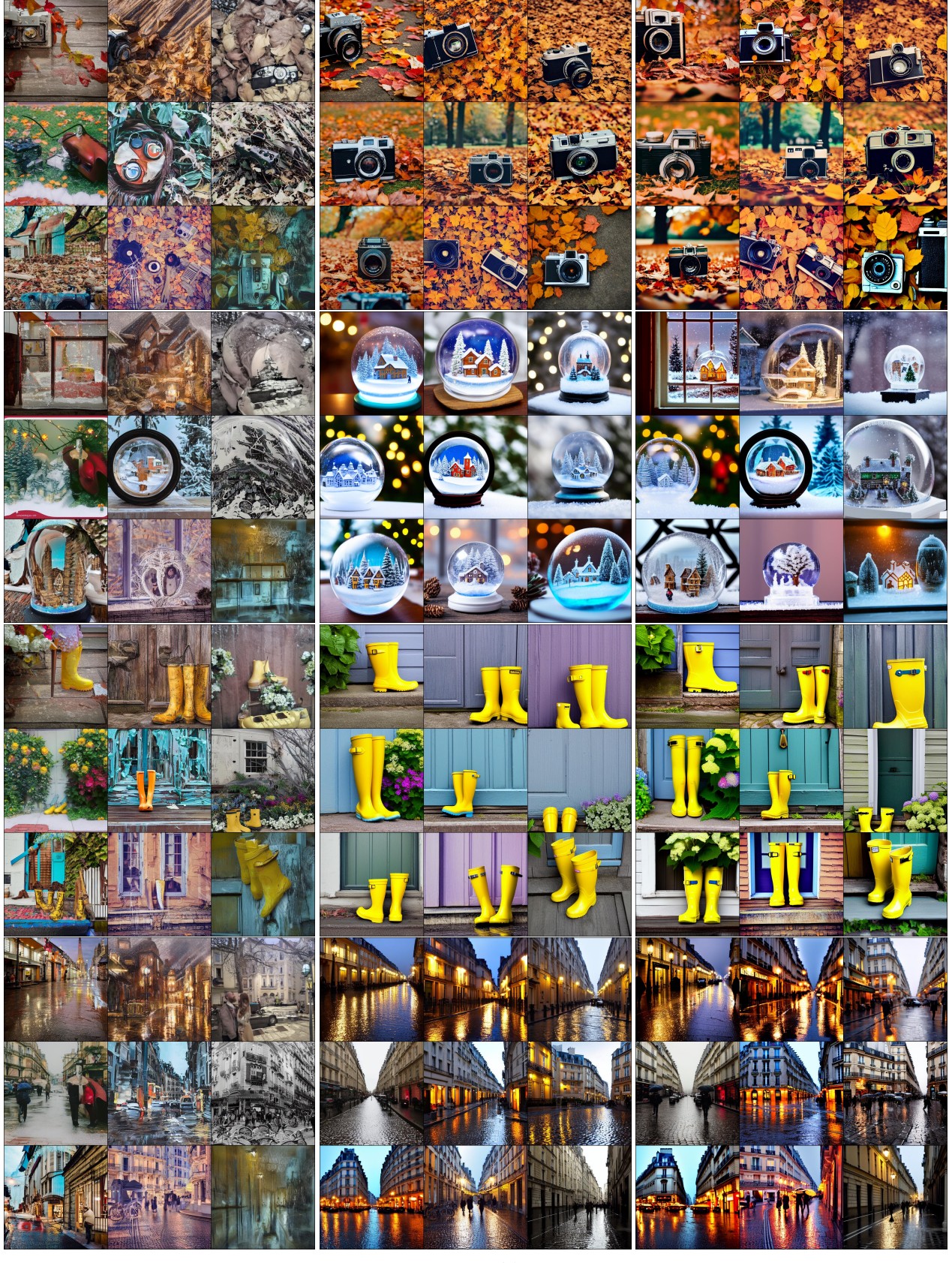

|  w/o Guidance | GFT (w/o Guidance) | w/ CFG Guidance |

Figure 14: Additional results of qualitative T2I comparison between vanilla conditional generation, GFT, and CFG on Stable Diffusion 1.5.

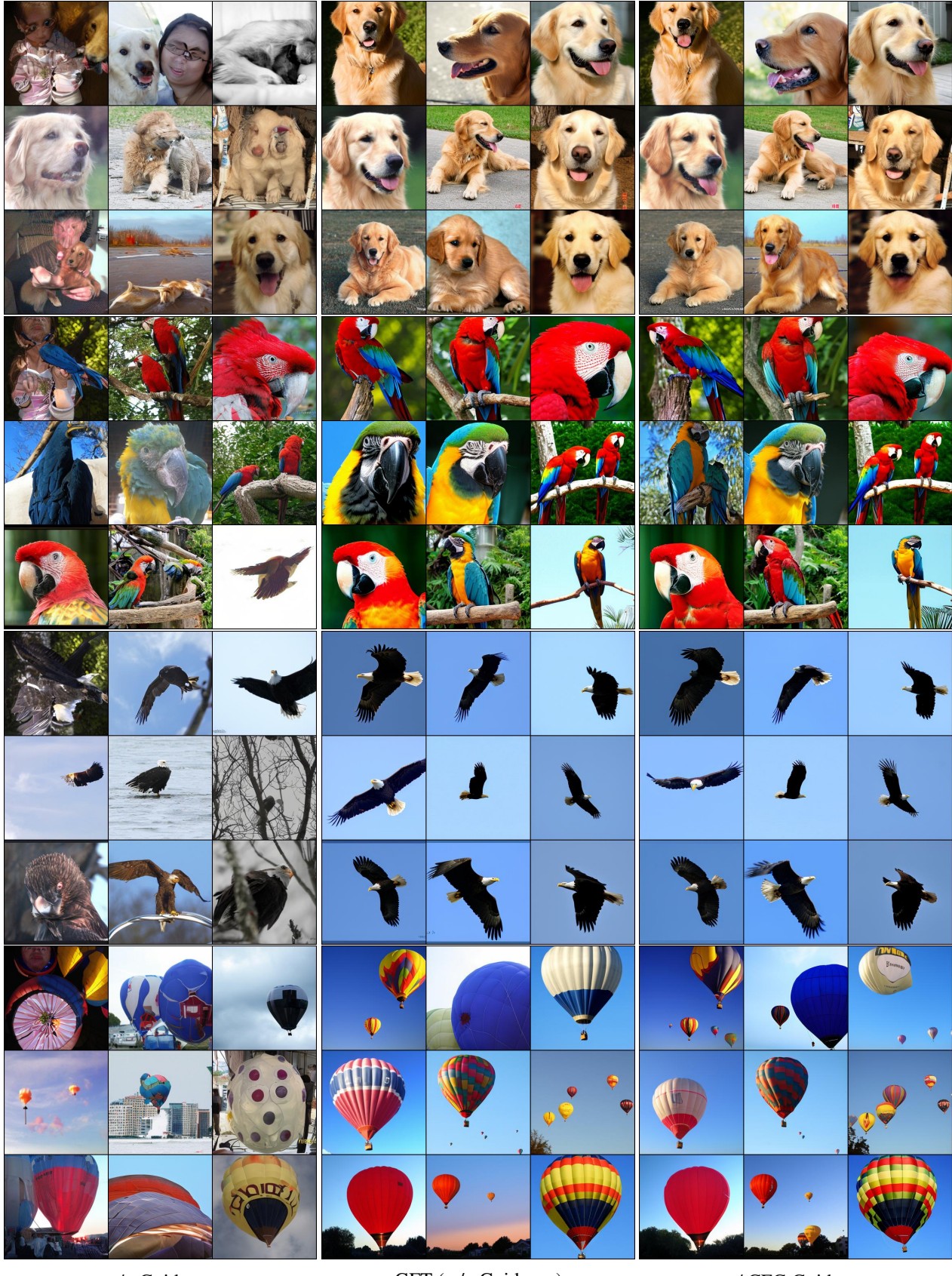

w/o Guidance  GFT (w/o Guidance)  w/ CFG Guidance

Figure 15: Additional results of qualitative C2I comparison between vanilla conditional generation, GFT, and CFG on DiT-XL/2.

# D. Implementation Details.

For all models, we keep training hyperparameters and other design choices consistent with their official codebases if not otherwise stated. We employ a mix of H100, A100 and A800 GPU cards for experimentation.

**DiT.** We mainly apply GFT to fine-tune DiT-XL/2 (28 epochs, 2% of pretraining epochs) and train DiT-B/2 from scratch (80 epochs, following the original DiT paper's settings (Peebles & Xie, 2023)). Since the DiT-B/2 pretraining checkpoint is not publicly available, we reproduce its pretraining experiment. For all experiments, we use a batch size of 256 and a learning rate of $1e-4$. For DiT-XL/2 fine-tuning experiments, we employ a cosine-decay learning rate scheduler.

For comparison, we also fine-tune DiT-XL/2 using guidance distillation, with a scale range from 1 to 5, while keeping all other hyperparameters aligned with GFT.

The original DiT uses the old-fashioned DDPM (Ho et al., 2020) which learns both the mean and variance, while GFT is only concerned about the mean. We therefore abandon the variance output channels and related losses during training and switch to the Dpm-solver++ (Lu et al., 2022) sampler with 50 steps at inference. For reference, our baseline, DiT-XL/2 with CFG, achieves an FID of 2.11 using DPM-solver++, compared with 2.27 reported in the original paper. All results are evaluated with EMA models. The EMA decay rate is set to 0.9999.

**VAR.** We mainly apply GFT to fine-tune VAR-d30 models (15 epochs) or train VAR-d16 models from scratch (200 epochs). Batch size is 768. The initial learning rate is $1e-5$ in fine-tuning experiments and $1e-4$ in pretraining experiments. Following VAR (Tian et al., 2024), we employ a learning rate scheduler including a warmup and a linear decay process (minimal is 1% of the initial).

VAR by default adopts a pyramid CFG technique on predicted logits. The guidance scale 0 decreases linearly during the decoding process. Specifically, let n be the current decoding step index, and N be the total steps. The $n$-step guidance scale $s_n$ is

$$s_n = \frac{n}{N-1} s_0.$$

We find pyramid CFG is crucial to an ideal performance of VAR, and thus design a similar pyramid $\beta$ schedule during training:

$$\beta_n = \left[ (\frac{n}{N-1})^\alpha (\frac{1}{\beta_0} - 1) + 1 \right]^{-1},$$

where $\beta_n$ represents the token-specific $\beta$ value applied in the GFT AR loss (Eq. 12). $\alpha \geq 0$ is a hyperparameter to be tuned.

When $\alpha = 0$, we have $\beta_n = \beta_0$, standing for standard GFT. When $\alpha = 1.0$, we have $\frac{1}{\beta_n} - 1 = (\frac{n}{N-1})(\frac{1}{\beta_0} - 1)$, corresponding to the default pyramid CFG technique applied by VAR. In practice, we set $\alpha = 1.5$ in GFT training and find this slightly outperforms $\alpha = 1.0$.

**LlamaGen.** We mainly apply GFT to fine-tune LlamaGen-3B models (15 epochs) or train LlamaGen-L models from scratch (300 epochs). For fine-tuning, the batch size is 256, and the learning rate is $2e-4$. For pretraining, the batch size is 768, and the learning rate is $1e-4$. We adopt a cosine-decay learning rate scheduler in all experiments.

**MAR.** We apply GFT to MAR-B, including both fine-tuning (10 epochs) and training from scratch (800 epochs). We find the batch size crucial for MAR and use 2048 following the original paper. For fine-tuning, we employ a learning rate scheduler including a 5-epoch linear warmup to $8e-4$ and a cosine decay process to $1e-4$. For training from scratch, we employ a 100-epoch linear lr warmup to $8e-4$, followed by a constant lr schedule, which is the same configuration as the original MAR pretraining.

The original MAR follows the old-fashioned DDPM (Ho et al., 2020) which learns both the mean and variance, while GFT is only concerned about the mean. We therefore abandon the variance output channels and related losses during training and switch to the DDIM (Song et al., 2020a) sampler with 100 steps at inference. As the $\beta$ condition may not precisely capture the effects of the guidance scale after training, we tune the inference $\beta$ schedule to maximize the performance. Specifically, we adopt a power-cosine schedule

$$\beta_n = \left[ \frac{1 - \cos((n/(N-1))^\alpha \pi)}{2} (\frac{1}{\beta_0} - 1) + 1 \right]^{-1}$$

where we choose $\alpha = 0.4$.

**Stable Diffusion 1.5.** We apply GFT to fine-tune Stable Diffusion 1.5 (SD1.5) (Rombach et al., 2022) for 70,000 gradient updates with a batch size of 256 and constant learning rate of $1e - 5$ with 1,000 warmup steps. We disable conditioning dropout as we find it improves CLIP score. For comparison, we also fine-tune SD1.5 using guidance distillation with a scale range from 1 to 14, while keeping other hyperparameters aligned with GFT.

For evaluation, following GigaGAN (Kang et al., 2023) and DMD (Yin et al., 2024), we generate images using 30K prompts from the COCO2014 (Lin et al., 2014) validation set, downsample them to 256×256, and compare with 40,504 real images from the same validation set. We use clean-FID (Parmar et al., 2022) to calculate FID and OpenCLIP-G (Ilharco et al., 2021; Cherti et al., 2023) to calculate CLIP score (Radford et al., 2021). All results are evaluated using 50 steps DPM-solver++ (Lu et al., 2022) with EMA models. The EMA decay rate is set to 0.9999.

# E. Prompts for Figure 14

We use the following prompts for Figure 14.

- A vintage camera in a park, autumn leaves scattered around it.

- Pristine snow globe showing a winter village scene, sitting on a frost-covered pine windowsill at dawn.

- Vibrant yellow rain boots standing by a cottage door, fresh raindrops dripping from blooming hydrangeas.

- Rain-soaked Parisian streets at twilight.

