# OpenReview forum: "Visual Generation Without Guidance"
_ICML.cc/2025/Conference — ICML 2025 poster_

### Official Review · Reviewer_2PrL · 2025-03-10

**Overall Recommendation:** 4

**Summary:**

1. They proposed guidance-free training, which reparameterizes the conditional model as a combination of trainable sampling model and frozen unconditional model.

2. They introduced pseudo-temperature input (β) to control the fidelity-diversity trade-off.

3. They reached similar performance compared to CFG across several tasks.

**Claims And Evidence:**

1. The quantitative results demonstrate that their method achieves comparable FID or even better FID than CFG.

2. The qualitative results show similar synthetic images compared with CFG.

3. Figure 3 demonstrates their computational efficiency.

**Essential References Not Discussed:**

No

**Experimental Designs Or Analyses:**

1. Fine-tuning efficiency experiments shown in Figure 5 track FID over epochs, demonstrating rapid convergence with minimal training.

2. The diversity-fidelity tradeoff experiments shown in Figures 7 and 8 vary the temperature parameter, providing a fair comparison with guided approaches.

**Methods And Evaluation Criteria:**

1. The evaluation metrics are insufficient, as the paper does not report sFID, precision, or recall. Those metrics were used in the original DiT experiments to assess fidelity and diversity.

2. The method is evaluated across various models, demonstrating its versatility and general applicability.

**Other Comments Or Suggestions:**

No

**Other Strengths And Weaknesses:**

Strength:
1. Compared to distillation methods, they can train from scratch.

2. The performance superior is convincible across many of different tasks.

**Questions For Authors:**

Can you report more comprehensive evaluation metrics to better support your claims?

**Relation To Broader Scientific Literature:**

GFT matches the performance of CFG while using half-time cost during inference.

**Theoretical Claims:**

Theorem 1 is mathematically proven in Appendix B, and shows GFT models the same sampling distribution as CFG.

---

> ### Author Rebuttal · Authors · 2025-03-30
>
> # Official Response to Reviewer 2PrL  (Part 1/1)
> We are glad the reviewer finds our method to be versatile and generally applicable. This is exactly what we are trying to convey in this paper.
> **Q1: More comprehensive evaluation metrics like sFID, precision, Recall**
>
> **A1:**
> We have conducted additional evaluation, and now report their full evaluation metrics. These results are consistent with our main findings and further support the effectiveness of GFT.
>
> **For fine-tuning experiments**, GFT consistently achieves better or comparable performance across all metrics, as shown below:
>
> DiT-XL/2
>
> | Method | FID | sFID | IS | Presicion | Recall |
> |:----:|:---:|:--:|:--:|:---:|:--:|
> | CFG | 2.11  | 4.81 | 245.7 | **0.81** | 0.59 |
> | GFT | **1.99** | **4.67** | **266.6** | **0.81** | **0.60** |
>
> LlamaGen-3B
>
> | Method | FID | sFID | IS | Presicion | Recall |
> |:----:|:---:|:---:|:---:|:--:|:---:|
> | CFG | 2.22  | 6.01 | 264.1 | 0.82 | 0.58 |
> | GFT | **2.21** | **5.85** | **279.0** | **0.83** | **0.59** |
>
> **For pretraining experiments**, GFT also shows consistent improvement, especially in FID, sFID, and IS:
>
> DiT-B/2
>
> | Method | FID | sFID | IS | Presicion | Recall |
> |:----:|:---:|:----:|:----:|:----:|:---:|
> | CFG | 9.72  | 8.75 | 161.5 | 0.84 | **0.34** |
> | GFT | **9.04** | **8.29** | **166.6** | **0.86** | **0.34** |
>
> LlamaGen-L
>
> | Method | FID | sFID | IS | Presicion | Recall |
> |:--:|:-:|:-:|:-:|:-:|:-:|
> | CFG | 3.06  | 6.15 | 257.1 | **0.83** | 0.52 |
> | GFT | **2.52** | **5.98** | **269.5** | 0.82 | **0.57** |
>
> More detailed numbers & visualizations at:
> https://anonymous.4open.science/r/Additional-Results-4CDD/README.md
> ***
> Below, we borrow some space to further respond to **Reviewer 5kGx** due to space limit.  Thank you for understanding!
> ***
> # Official Response to Reviewer 5kGx  (Part 2/2)
>
> ## Additional Input of beta
>
> **Q7: GFT saves much compute ..... however it comes at a cost. The guidance hyper-parameter becomes part of the model, which complicates the training of the model.**
>
> **A7:**
> We want to offer the reviewer a new perspective for incorporating $\beta$ as the model's input.
>
> 1. $\beta$ does not complicate **training**, because it is merely a **sampling** hyperparameter. In training, $\beta$ is always sampled from the uniform distribution [0,1] and does not need to be "tuned" (Algorithm 1 in paper). Thus **$\beta$ is very like diffusion time-step $t$**, which also serves as a scalar input of the model, but no one should think input $t$ is a "cost" or "training complication".
>
> 2. $\beta$ needs to be adjusted for GFT during **sampling**. However, guidance scale $s$ also needs to be tuned for CFG. So, we can conclude **GFT is no more complicated than CFG** during inference. Plus, GFT is 2x more efficient.
>
> **Q8: The hyper-parameter beta now becomes part of the model, ..., However, this makes the optimization hard as one needs to sample different values of beta.**
>
> **A8:** We are glad the reviewer mentions this concern, because **our experiments exactly prove "sampling different values of beta" does NOT complicate optimization**.
>
> The most clear evidence is Figure 6 (convergence plot) in the paper. In large-scale from-scratch training, the losses for GFT converge as fast as the classic CFG training w/o a $\beta$ input.
>
> Also in Table 4, performance numbers across 3 distinctive models clearly show that GFT w/ $\beta$ as model input slightly outperforms CFG baselines.
>
> **Q9: I checked the proof yet was not fully convinced. ... beta is part of the target model and can be adjusted .... Why is beta set to be 1 in the proof?**
>
> **A9:**
> The reviewer is referring to the proof for the unconditional model convergence point in Line 730-736 in Appendix B, where we select $\beta = 1$ to derive the conclusion.
>
> As a matter of fact, for other $\beta < 1$, the convergence point is the same. **We have already proved this point** in the following lines 737-748. At line 749, we summarized: "... this does not change the convergence point of loss. The optimal unconditional solution remains the same".
>
> We thank the reviewer for the comment and have updated our proof for better clarity [1].
>
> ## Formatting and clarity
>
> **Q10: Some notations are used without any proper definitions, such as $p^s$.**
>
> **A10:**
>
> We had explicitly defined $p^s$ in the background section $2.2$, Eq. 5, and referred to this definition in the Method section $3.1$, Line 141.
>
> In short, $p^s$ corresponds to our target sampling distribution. $p^s_\theta$ is our sampling model.
>
> $
>     p^\text{s}(x|c) \propto p(x|c) \left[\frac{p(x|c)}{p(x)}\right]^s.
> $
>
> **Q11: The loss forms in Table 1 are not reasonable....  Suggest revising them and providing more clarification.**
>
> **A11:**
> We thank the reviewer for the suggestion. Previously, we abbreviated some terms in Table 1 due to space constraints. We have now updated Table 1 to focus more on equation clarity and linked it to the formal equation in the paper [1].

---

> > ### Comment · Reviewer_2PrL · 2025-04-04
> >
> > I maintain my rate as Accept. The rebuttal addressed my concern regarding GFT achieving better results than CFG on sFID, IS, precision, and recall.

---

### Official Review · Reviewer_DEzb · 2025-03-13

**Overall Recommendation:** 3

**Summary:**

This paper introduces Guidance-Free Training (GFT), a novel method for training visual generative models that eliminates the need for Classifier-Free Guidance (CFG) during inference while maintaining comparable generation quality. The key insight is to do direct optimization of the desired sampling distribution during training by replacing the prediction with the linear interpolation with actual CFG target. GFT achieves comparable FID scores to CFG across multiple visual models.

**Claims And Evidence:**

The paper's claims about computational efficiency and model performance are well-supported by extensive experiments across five different model architectures.

**Essential References Not Discussed:**

NA

**Experimental Designs Or Analyses:**

The experimental design is sound, covering both fine-tuning and from-scratch training scenarios across multiple architectures.
The analysis of training dynamics (Fig. 6) and temperature control (Fig. 2) effectively supports the method's stability and flexibility.

**Methods And Evaluation Criteria:**

The evaluation is appropriate, using standard metrics (FID, IS) and datasets (ImageNet, COCO) that are widely accepted in the field of visual generation. The comparison with state-of-the-art techniques like CFG, guidance distillation, and contrastive alignment provides a comprehensive assessment.

**Other Comments Or Suggestions:**

* Consider providing more intuition for how to select $\beta$ values for both training and inference.
* The paper could benefit from a more detailed analysis of when GFT might not be advantageous compared to CFG.

**Other Strengths And Weaknesses:**

**Strengths**
* A method application to different visual generation models.
* Practical value in reducing inference computation by 50% eliminating the need of a unconditional inference in CFG.
* Simple implementation requiring minimal code changes to existing models


** Weaknesses**
* Introduces an additional hyperparameter $\beta$ that requires tuning in training and still in inference.
* $\beta$ is an input condition to the model, this may introduce varies issues of on how to inject this condition into the model. However there lacks ablation study on this.
* Limited exploration of how the approach extends to text-to-image generative models.

**Questions For Authors:**

1. How would GFT perform on more complex generation tasks such as text to image generation?
2. The paper mentions a 10-20% increase in training computation. How does this trade-off change with model scale?
3. Recently it is found that CFG doesn't work well with semantic latent space [1], does CFT present any advantage over CFG with such semantic tokenizers?

[1] Masked Autoencoders Are Effective Tokenizers for Diffusion Models.

**Relation To Broader Scientific Literature:**

This paper addresses a inefficiency issue in current visual generation pipelines by eliminating CFG linear inference of both conditional models and unconditional models.

**Theoretical Claims:**

I verified the correctness of Theorem 1, which provides the optimal solution for GFT. The proof in Appendix B logically demonstrates that stopping the unconditional gradient does not change the convergence point of the objective function.

---

> ### Author Rebuttal · Authors · 2025-04-01
>
> # Official Response to Reviewer DEzb
>
> **Q1: GFT introduces an additional hyperparameter $\beta$ that requires tuning in training and still in inference.**
>
> **A1:**
> We believe there is some misunderstanding over how we tune and inject $\beta$.
>
> 1. $\beta$ is not a training hyperparameter and **it does not require tuning during training**.
>
> (Algorithm 1): GFT is essentially learning various sampling models under different $\beta$ **at the same time**.
>
> $\beta$ is always randomly sampled from the uniform distribution [0,1] for each data point.
>
> **$\beta$ is very like the diffusion time-step $t$**, it is only a scalar model input instead of a hyperparameter to be tuned.
>
> 2. $\beta$ indeed needs to be adjusted in **sampling**. However, guidance $s$ also needs to be tuned for CFG. So,  **GFT requires no more tuning than CFG**.
>
> **Q2: Providing more intuition for how to select values for both training and inference.**
>
> **A2:** Following **A1**, we do not "select" $\beta$ in inference.
>
> In sampling, according to Theorem 1 and Line 192 in paper,  there exists a one-to-one correspondence between CFG $s$ and GFT $\beta$.
>
> $\beta = \frac{1}{1+s}$.
>
> Suppose we already know optimal CFG $s$ is 0.4, then the optimal GFT $\beta$ should be $\frac{1}{1.4}$. In practice, this value is usually not accurate due to training bias, but it provides a good starting point.
>
> CFG:
>
> | Guidance Scale $s$ |FID|IS|Guidance-free?|
> |:-:|:-:|:-:|:-:|
> | 0.0|9.34|117.1|Yes|
> |0.35 |2.22 | 230.8| No|
> |**0.4**| **2.11**| 245.7|No|
> |0.45|2.14|258.6| No|
> |0.5|2.14| 271.2 | No|
>
> GFT:
>
> | Beta $\beta$| FID| IS | Guidance-free? |
> |:-:|:--:|:--:|:--:|
> |1.0|6.77|152.8|Yes|
> |1/1.35 | 2.29 | 203.5 | Yes |
> |**1/1.4** | **2.07** | 229.7 | Yes |
> |1/1.45|1.99|240.0| Yes |
> |1/1.5| 1.99|249.6| Yes |
>
>
> **Q3: $\beta$ may introduce various issues in how to inject this condition. However there lacks ablation study on this.**
>
> **A3:**
> We are happy the reviewer is interested in this.
>
> Actually, how to inject a scalar condition into a diffusion model is well-explored, because diffusion-time $t$ is similar stuff. We simply borrowed their design and found it works well enough.
>
> We indeed tried several ablation choices initially.
>
> 1. Instead of posing $\beta$ as model input, we leverage $s = \frac{1}{\beta} -1$ in Theorem 1, and model
>
> $
> \epsilon\_\theta(x\_t|c, \beta) := \epsilon\_\theta^1(x\_t|c) + (\frac{1}{\beta} - 1) \epsilon\_\theta^2(x\_t|c),
> $
>
> where $\epsilon\_\theta^1$ is the pretrained network, and $\epsilon\_\theta^2$ the pretrained network with a new MLP head. Our hope is to avoid making $\beta$ as network input. However, this design performs poorly:
>
> | Model | CFG FID| $\beta$ as input FID|$\beta$ **as linear coefficient FID** |
> |:--:|:--:|:--:|:--:|
> | LlamaGen-3B | 2.22|**2.21** | 2.44 |
> | VAR-d30 | 1.92|**1.91** | 2.30 |
>
> We suspect the main reason is that '$\beta$ as model input' allows better leveraging the full potential of pretrained model parameters.
>
> 2. We ablated how the $\beta$ input MLP encoder size on DiT-XL. (Turns out to be insensitive)
>
> | MLP encoder layers | FID |
> |:-:|:--:|
> | 1 | 1.93 |
> | 2 | 1.92 |
> | 3 | 1.92 |
>
> **Q4: Limited exploration of how the approach extends to text-to-image generative models.**
>
> **A4:** We feel the reviewer might miss Table 3, Figure 4, Figure 8, and Figure 12 in our paper, where we have **already conducted experiments on T2I generative models** using Stable Diffusion as base model, and LAION 5+ as the dataset.
>
> In short, GFT significantly increases guidance-free FID from 22.55 to 8.10, CLIP score from 0.252 to 0.313, achieving same level of CFG performance.
>
> **Q5: When GFT might not be advantageous compared to CFG?**
>
> **A5:**
>
> 1. If training from scratch, GFT still requires  20\% more computation than CFG, but is 2x more efficient in inference.
>
> 2. When it comes to advanced guidance methods, such as dynamically adjusting the guidance scale during decoding [1]. CFG only requires modifying the sampling code. However, GFT requires redesigning the training code (though only 1-3 lines).
>
> [1] Applying guidance in a limited interval improves sample quality in diffusion models.
>
> **Q6: How does CFG/GFT training computation trade-off change with model scale?**
>
> **A6:** **In short, the influence of model size is almost negligible.**
>
> VAR as an example:
>
> |Model|Size|Batch Size|Acc. step|Time/Epoch (CFG)|Time/Epoch (GFT)| Time (GFT/CFG)|
> |--|--|--|--|---|---|--|
> |VAR-d16|300M|768|1|0.82h|0.93h|1.134|
> |VAR-d20|600M|768|4|1.11h|1.26h|1.135|
> |VAR-d24|1B|768|4|1.33h|1.52h|1.142|
> |VAR-d30|2B|768|12|2.07h|2.37h|1.145|
>
> **Q7: CFG doesn't work well with semantic latent space. Does GFT present any advantage over CFG with such semantic tokenizers?**
>
> **A7:**
> Unfortunately, due to the similar theoretical property between GFT and CFG  as in Theorem 1, we find it difficult to expect GFT to solve some potential issues in which CFG has failed. Likewise, if GFT does not work well on some domains, we believe CFG may also struggle.

---

### Official Review · Reviewer_oBo4 · 2025-03-17

**Overall Recommendation:** 4

**Summary:**

* “Visual Generation Without Guidance” presents Guidance-Free Training (GFT), a novel approach for visual generative models that aims to eliminate the need for guided sampling and reduce computational costs.

* The core of GFT design is to transform the target sampling model into an easily learnable form. Instead of explicitly learning a conditional network as in CFG, GFT defines the conditional model implicitly as a linear interpolation of a sampling network and an unconditional network. During training, GFT optimizes the same conditional objective as CFG.

* GFT is an alternative to guided sampling in visual generative models. It achieves comparable performance to CFG while reducing sampling computational costs by 50%. The method is simple to implement, requiring minimal modifications to existing codebases, and can be trained directly from scratch. It represents an advancement in making high - quality visual generation more efficient and accessible.

**Claims And Evidence:**

The claims made in the submission are clear

**Essential References Not Discussed:**

Essentially, all related works have been properly cited and discussed in the paper.

**Experimental Designs Or Analyses:**

Yes, I checked the soundness and validity of the experimental designs and analyses. GFT conducted experiments on the baselines of many mainstream methods(DIT, VAR, MAR, LlamaGen) and validated the results across multiple benchmarks. The experimental design and analysis are reasonable.

**Methods And Evaluation Criteria:**

The methods and evaluation criteria make sense for the problem.

**Other Comments Or Suggestions:**

refer to "Other Strengths And Weaknesses"

**Other Strengths And Weaknesses:**

After thoroughly reading the paper and following the derivations of the formulas, I find this paper significantly meaningful and quite enjoyable. I believe that the classifier-free guidance (CFG) technique in visual generation will inevitably be replaced in the future, and this paper theoretically demonstrates the feasibility of this transition.

The paper is well-written, the experiments are comprehensive(especially the experiment for FID-IS trade off experiments), and the results are demonstrated across a substantial number of related works. However, I have a few minor questions:

1. The paper mentions the stop gradient part, and I am very curious about what impact removing the stop gradient would have on performance.
2. I am also curious about the effect that the size of the MLP model following the β parameter has on the results.
3. If it's a T2I task, how should the negative prompt and  β be implemented?

**Questions For Authors:**

refer to "Other Strengths And Weaknesses"

**Relation To Broader Scientific Literature:**

I believe this paper significantly contributes to the acceleration and improvement of generative models. Additionally, I think that the classifier-free guidance (CFG) technique in visual generation will inevitably be replaced in the future.

**Theoretical Claims:**

Yes, I  check the correctness of most proofs for theoretical claims.

In line 8 of Algorithm 1 in the paper, (c_{\varnothing}= c) masked by (\varnothing) with a 10% probability, I am a bit confused. Given that we already have the pseudo-temperature (\beta) ~ ( (0, 1) ), why do we still need to perform dropout on the condition? When dropping out the condition, wouldn't the two branches ((\beta) and (1-\beta)) be redundant?

---

> ### Author Rebuttal · Authors · 2025-03-31
>
> # Official Response to Reviewer oBo4 (Part 1/1)
> We thank the reviewer for the insightful review. We are really glad to see the reviewer shares the same belief as us that Guided sampling should eventually be removed from visual modeling. We are also greatly motivated by the high praise given to our work.
>
> **Q1: Algorithm 1, line 8: We already have the pseudo-temperature $\beta \in [0,1]$, why do we still need to perform dropout on the condition?**
>
> **A1:** The most direct reason for dropping out condition is that we have detached the gradient from unconditional model ($\mathrm{\mathbf{sg}}(\cdot)$) during training (Algorithm 1 line 12). If we do not randomly dropout the condition, the unconditional model would not be trained at all.
>
> $L_\theta
>        =\|\beta  \epsilon\_\theta^s(x\_t|c\_\emptyset,\beta ) + (1-\beta) \mathrm{\mathbf{sg}}\[ \epsilon\_\theta^u (x\_t| c=\emptyset , \beta=1)\] - \epsilon\|^2.
> $
>
> The question now becomes why we have to detach the unconditional gradient.
>
> In short, without detaching the unconditional gradient, loss becomes
>
> $L_\theta
>        =\|\beta  \epsilon\_\theta^s(x\_t|c,\beta ) + (1-\beta) \[ \epsilon\_\theta^u (x\_t| c=\emptyset , \beta=1)\] - \epsilon\|^2.
> $
>
> Since $\beta \sim \text{U}[0,1]$, the unconditional model can still be trained. However, the loss function now spends 50 \% of its energy to optimize the unconditional model, which is way too much because the unconditional part is not what we finally want. **This hurts performance in practice.** Also, this causes misalignment with CFG training pipeline. **We believe GFT should not only ensure soundness but also offer seamless integration and extreme compatibility.**
>
> We refer the reviewer to Section 3.2 (lines 194 to 210) in our paper for detailed response.
>
>
> **Q2: When dropping out the condition, wouldn't the two branches $\beta$ and$ (1-\beta)$ be redundant?**
>
> **A2:**
> Yes, they will be redundant. However, when dropping out condition, we are training the unconditional model, the loss is
>
> $L_\theta
>        =\|\beta  \epsilon\_\theta^u(x\_t|\beta ) + (1-\beta)  \mathrm{\mathbf{sg}} \[ \epsilon\_\theta^u (x\_t| \beta)\] - \epsilon\|^2.
> $
>
> We have proved in Appendix B (line 737-748) that this objective has exactly the same solution as classic unconditional loss:
> $L_\theta
>        =\| \epsilon\_\theta^u(x\_t|\beta )  - \epsilon\|^2.
> $
>
>
> Therefore, $\mathbf{sg}$ does not affect model convergence.
>
> We could have removed the $1-\beta$ branch when dropping out condition, but again, this causes the unconditional model part to be trained too much and also breaks compatibility with CFG design. We considered and tried many versions of this loss equation, but decided the first loss form is the most **elegant** one.
>
>
> **Q3: I am very curious about what impact removing the stop gradient would have on performance.**
>
> **A3:**
> We re-ran the DiT-XL/2 experiments following the settings in the paper, with and without stop-gradient. The results show that using stop-gradient slightly improves the performance.
>
> | stop gradient? | FID |
> |:-------:|:-----------------:|
> | Yes | 1.93 |
> | No | 2.04 |
>
> **Q4: I am also curious about the effect that the size of the MLP model following the $\beta$ parameter has on the results.**
>
> **A4:** We ablated how the $\beta$ MLP encoder size affects training performance on DiT-XL/2. Overall, we find GFT to be insensitive to the MLP encoder size:
>
> | MLP layers | FID |
> |:-------:|:-----------------:|
> | 1 | 1.93 |
> | 2 | 1.92 |
> | 3 | 1.92 |
>
> **Q5: If it's a T2I task, how should the negative prompt and β be implemented?**
>
> **A5:**
>
> 1. Replace the unconditional mask with a negative prompt $c\_n$ randomly sampled from a pool of negative candidates.
> 2. We assume randomly masking out "condition (prompt)" might not be necessary anymore. Because these negative prompts should already appear in the dataset several times. We are not sure, this one can be tested out.
>
> $L_\theta
>        =\|\beta  \epsilon\_\theta^s(x\_t|c,\beta ) + (1-\beta) \mathrm{\mathbf{sg}}\[ \epsilon\_\theta^s (x\_t| c\_n , \beta=1)\] - \epsilon\|^2.
> $

---

> > ### Comment · Reviewer_oBo4 · 2025-04-04
> >
> > After reading the rebuttal, I maintain my rate as Accept.

---

### Official Review · Reviewer_5kGx · 2025-03-17

**Overall Recommendation:** 3

**Summary:**

In this work the authors have proposed a technical to improve the standard classifier-free guidance. The key idea is to directly optimize the target guided noise regressor (as described in Eq. 6) after converting Eq. 4 into another form amenable for this purpose. The standard CFG demands running the diffusion model twice, with or without prompt separately. The proposed GFT method incapsulates all compute into a single model, thus saving much computations.

## update after rebuttal

My previous concerns have been mostly addressed in a convincing way. Thus I will raise my score to be weak accept.

**Claims And Evidence:**

The major claims include the efficacy and effectiveness of the proposed GFT method, compared with classifier-free guidance and two existing relevant baselines (guidance distillation and condition contrastive alignment). The authors provide empirical evaluations on several widely-used benchmarks. However, the reported experimental results are not consistent in all tables or figures. For example, from Figure 7, it seems by properly tuning some hyper-parameters (in particular, the diffusion temperature related s or beta) the proposed method and baselines perform similarly. But Tables 2 and 3 report a clear, large margin between GFT and others.

**Essential References Not Discussed:**

The references are sufficient.

**Experimental Designs Or Analyses:**

1. Some reported results are superficially treated. For example, the authors did not specify the basis for comparing the training time and GPU memory usage in a convincing way. In Section 5.2, it was superficially claimed that "less than 5% pretraining computation" and "being 2x faster in sampling". The comparison should be in more rigorous setting.

2. The claim is somewhat counting intuition. Eq. 6 is essentially the same to Eq. 4. The major difference is to set epsilon^s directly to be optimized such that much computation can be saved. The authors claim that such a reformulation can bring notable performance gain as shows in Tables 2 and 3, which needs further clarification.

**Methods And Evaluation Criteria:**

The evaluations are conducted following previous practice (class-conditional image generation on ImageNet and text-to-image generation on LAION). All dataset and metrics are reasonable.

**Other Comments Or Suggestions:**

There are some ad hoc parameters in the proposed model. For example, in Algorithm 1, it was shown that c will be masked with a 10% change. How was this 10% chosen? Any specific reason for such a parameter?

Some notations are used without any proper definitions, such as p^s.

The loss forms in Table 1 are not reasonable, in particular the loss for guidance-free training. The authors are suggested to revise them and provide more clarification.

**Other Strengths And Weaknesses:**

Improving classifier guidance is an important research topic in AIGC. I am surprised that the simple change in the proposed method can bring much improvement in the experiments. However, the experiments in their current form are not fully convincing to me, which makes me hesitate to recommend acceptance.

In other sections of the reviews, I have discussed about several potential weaknesses or problems in the work, including the proof of theorem 1, the claimed superiority in terms of FID. There are some other issues that I feel to be critical for the final evaluation of this work. First, the hyper-parameter beta now becomes part of the model, such as one can directly adjust beta for obtaining different levels of guidance. However, this makes the optimization hard as one needs to sample different values of beta (if I understand well). Also very importantly, in comparing the proposed method and other baselines, it is critical to ensure that they are on the same level of guidance strength (i.e., with proper s and beta), such that a fair comparison can be guaranteed. However, this is missing in many experiments, such as the one in Figure 5.

The reformulation makes there are two different models active and up to optimization during training, namely the s-type and u-type models. Are they sharing same parameters as in standard classifier-free guidance? If not, that makes the advantages of GFT less obvious.

Figure 2 is not very informative since most related works can demonstrate such evolution under stronger level of guidance.

**Questions For Authors:**

In the training stage, are there two models (one with beta, and the other the unconditional version) optimized in the proposed GFT, or just one? If the former case were true, does it require much more memory space in comparison with CFG, where only one model is kept, with the prompt turned on or off?

**Relation To Broader Scientific Literature:**

Essentially the proposed model is an incremental improvement to the standard classifier-free guidance. It saves much compute in the inference time (since only one model will be executed, rather than two as in CFG), however it comes at a cost. The guidance hyper-parameter becomes part of the model, which complicates the training of the model.

I regard the work to be interesting to a broad spectrum of readers, since CFG is crucial for diffusion based generation. However the proposed method needs further justification to be more convincing.

**Theoretical Claims:**

The authors provided proof for Theorem 1 in appendix B. I checked the proof yet was not fully convinced. According to my understanding, beta is part of the target model and can be adjusted for different strength of guidance. Why is beta set to be 1 in the proof?

---

> ### Author Rebuttal · Authors · 2025-03-31
>
> # Official Response to Reviewer 5kGx (Part 1/2)
> We thank the reviewer for the very detailed comments! We summarize concerns into four categories:
> ## Computational/Memory efficiency
> **Q1: The reformulation results in two different models active and up to optimization during training, ... Are they sharing the same parameters as in CFG? If not, that makes the advantages of GFT less obvious.**
>
> **A1:**
> **They do share parameters just like CFG**, Throughout all our experiments, there is ALWAYS only one model kept in memory. This makes GFT extremely memory-efficient.
>
> As noted in Figure 3, GFT has the **same** GPU memory usage as CFG training. This distinguishes GFT from all previous distillation approaches, which all require more GPU memory than CFG.
>
> LlamaGen exp: 8H100, bz 256, FSDP
> | Model | Memory Per Card | Ratio |
> |:-:|:-:|:-:|
> | CFG | 59.3G | 1.0000|
> | GFT | 59.4G | 1.0016|
>
> **Q2: Did not specify the basis for comparing the training time in a convincing way. ... superficially claimed "less than 5\% pretraining computation" and "2x faster in sampling" .... should be more rigorous.**
>
> **A2:**
> We thank the reviewer for the suggestion. We use VAR and DiT to show how "less than 5\% pretraining computation" is calculated in detail.
>
> (8*H100 GPUs)
>
> |Model|Size|Batch Size|Acc. step|Pretrain Epoch|Time/Epoch (CFG)|Total (CFG)|Finetune Epoch|Time/Epoch (GFT)|Total (GFT)|Ratio GFT/CFG|
> |--|-|-|-|-|-|-|-|--|--|--|
> |VAR-d30|2.0B|768|12|350|2.07h|724.5h|15|2.37h|35.55h|4.90\%|
> |DiT-XL|675M|256|1|1400|0.29h|406h|28|0.33h|9.24h|2.27\%|
>
> "2x faster in sampling": our method simply halves the model inference, ==> allows doubling the batch size while keeping the same sampling time.
>
> ## Method motivation and Evaluation
>
> **Q3: The reported experimental results are not consistent..... Figure 7 : GFT and baselines perform similarly..... But Tables 2/3 report a clear, large margin between GFT and CFG.**
>
> **A3:**
> We feel there is clearly some misunderstanding here. **Figure 7 and Tables 2/3 show consistent results, they just focus on different evaluation metrics.** Take DiT-XL for instance.
>
> CFG:
>
> | Guidance $s$ | FID | IS | Guidance-free? |
> |:----:|:---:|:----:|:---:|
> | 1.0 | **9.34** |117.1 | **Yes** (Table 2) |
> |1.35 | 2.22 | 230.8 | No |
> |1.4 | **2.11** | 245.7| **No** |
> |1.45 | 2.14 |258.6| No |
> |1.5 | 2.14 | 271.2 | No |
>
> GFT:
>
> | Beta $\beta$| FID | IS | Guidance-free? |
> |:--:|:-:|:--:|:--:|
> |1.0|6.77|152.8 | Yes |
> |1/1.35|2.29| 203.5 | Yes |
> |1/1.4|2.07| 229.7 | Yes |
> |1/1.45|1.99| 240.0 | Yes |
> | 1/1.5| **1.99** |249.6 |**Yes**|
>
> Two ways to interpret this data:
>
> 1. If restricted to **guidance-free** sampling,  ==> can only use $s=1$ for CFG.   ==>  GFT's FID  1.99 **significantly outperforms** CFG FID 9.34. ====> **Table 2/3**.
>
> 2. If only focus on the FID-IS trade-off.  ===> we can tune CFG $s$.  ==>  GFT (FID 1.99) slightly outperforms CFG (FID 2.11) , similar FID-IS trade-off.  ====> **Figure 7**
>
> **Q4: Figure 2 is not very informative since most related works can demonstrate such evolution under a stronger level of guidance.**
>
> **A4:** Following **A3**, Figure 2 does NOT mean to prove GFT "outperforms" existing methods like CFG. Instead, it demonstrates GFT can achieve previous work performance **without** guided sampling. Thus, the reviewer feeling "GFT is similar to related works with a strong level of guidance" is exactly what we want to see.
>
> We thank the reviewer's question and have updated Figure 2's caption to avoid possible misleading [1].
>
> [1] https://anonymous.4open.science/r/Additional-Results-4CDD/updated.pdf
>
> **Q5: The claim is somewhat counting intuition. Eq. 6 is essentially the same as Eq. 4... However, the authors claim GFT can bring notable performance gain as shown in Tables 2/3.**
>
> **A5:**
> We respectfully disagree with the reviewer. We hope **A3** and **A4** address the reviewer's concern. **
>
> 1.  Theoretically, GFT and CFG are equivalent.  (Eq. 4 -> Eq. 6)
> 2. **guidance-free** performance: GFT is significantly better. (Table 2/3)
> 3. FID-IS trade-off: GFT and CFG perform similarly. GFT is more efficient in sampling. (Figure 7)
>
> **Q6: In comparing GFT and other baselines, it is critical to ensure they are on the same level of guidance strength. However, this is missing ...**
>
> **A6:**
> We respectfully disagree. For GFT and all baselines, we report their **best** performance by tuning their respective guidance $s$ or temperature $\beta$. This ensures fairness.
>
> If have to align guidance strength, we can easily select a hyperparameter suitable for GFT but not optimal for CFG. For example, for DiT-XL, GFT achieves optimal FID 1.99 with $\beta = 1/1.5 = 0.667$. Under the same level of guidance, CFG FID is **2.14** at $s=1.5$. However, we choose to compare with the CFG optimal FID **2.11** at $s=1.4$.
> ***
> # Reminder for Part (2/2)
> Due to very detailed questions posted, we borrowed some space from **Reviewer 2PrL** for **Part (2/2)** response to answer **Q7-Q11**.  Thank you for understanding!

---

### Decision · Program_Chairs · 2025-05-01

**Decision:**

Accept (poster)

**Comment:**

The paper proposes Guidance-Free Training (GFT), a novel method that optimizes classifier-free guidance (CFG) more efficiently by reformulating the guidance mechanism to eliminate the need for guided sampling at inference. Reviewers appreciate the theoretical clarity, practical effectiveness, and the computational efficiency improvements (approximately 2x faster inference) demonstrated on standard benchmarks. The authors also effectively addressed the concerns raised by the reviewers during rebuttal. Given the positive consensus among reviewers, I recommend acceptance for this paper.